# Learning Overcomplete HMMs

**Vatsal Sharan**
Stanford University
vsharan@stanford.edu

**Sham Kakade**
University of Washington
sham@cs.washington.edu

**Percy Liang**
Stanford University
pliang@cs.stanford.edu

**Gregory Valiant**
Stanford University
valiant@stanford.edu

## Abstract

We study the problem of learning overcomplete HMMs—those that have many hidden states but a small output alphabet. Despite having significant practical importance, such HMMs are poorly understood with no known positive or negative results for efficient learning. In this paper, we present several new results—both positive and negative—which help define the boundaries between the tractable and intractable settings. Specifically, we show positive results for a large subclass of HMMs whose transition matrices are sparse, well-conditioned, and have small probability mass on short cycles. On the other hand, we show that learning is impossible given only a polynomial number of samples for HMMs with a small output alphabet and whose transition matrices are random regular graphs with large degree. We also discuss these results in the context of learning HMMs which can capture long-term dependencies.

## 1   Introduction

Hidden Markov Models (HMMs) are commonly used for data with natural sequential structure (e.g., speech, language, video). This paper focuses on *overcomplete* HMMs, where the number of output symbols $m$ is much smaller than the number of hidden states $n$. As an example, for an HMM that outputs natural language documents one character at a time, the number of characters $m$ is quite small, but the number of hidden states $n$ would need to be very large to encode the rich syntactic, semantic, and discourse structure of the document.

Most algorithms for learning HMMs with provable guarantees assume the transition $T \in \mathbb{R}^{n \times n}$ and observation $O \in \mathbb{R}^{m \times n}$ matrices are full rank [2, 3, 20] and hence do not apply to the overcomplete regime. A notable exception is the recent work of Huang et al. [14] who studied this setting where $m \ll n$ and showed that *generic* HMMs can be learned in polynomial time given *exact* moments of the output process (which requires infinite data). Though understanding properties of generic HMMs is an important first step, in reality, HMMs with a large number of hidden states typically have structured, non-generic transition matrices—e.g., consider sparse transition matrices or transition matrices of factorial HMMs [12]. Huang et al. [14] also assume access to exact moments, which leaves open the question of when learning is possible with efficient sample complexity. Summarizing, we are interested in the following questions:

1. What are the fundamental limitations for learning overcomplete HMMs?
2. What properties of HMMs make learning possible with polynomial samples?
3. Are there structured HMMs which can be learned in the overcomplete regime?

**Our contributions.** We make progress on all three questions in this work, sharpening our understanding of the boundary between tractable and intractable learning. We begin by stating a negative result, which perhaps explains some of the difficulty of obtaining strong learning guarantees in the overcomplete setting.

**Theorem 1.** *The parameters of HMMs where i) the transition matrix encodes a random walk on a regular graph on $n$ nodes with degree polynomial in $n$, ii) the output alphabet $m = polylog(n)$ and,*

*iii) the output distribution for each hidden state is chosen uniformly and independently at random, cannot be learned (even approximately) using polynomially many samples over any window length polynomial in $n$, with high probability over the choice of the observation matrix.*

Theorem 1 is somewhat surprising, as parameters of HMMs with such transition matrices can be easily learned in the non-overcomplete ($m \geq n$) regime. This is because such transition matrices are full-rank and their condition numbers are polynomial in $n$; hence spectral techniques such as Anandkumar et al. [3] can be applied. Theorem 1 is also fundamentally of a different nature as compared to lower bounds based on parity with noise reductions for HMMs [20], as ours is information-theoretic.[1] Also, it seems far more damning as the hard cases are seemingly innocuous classes such as random walks on dense graphs. The lower bound also shows that analyzing generic or random HMMs might not be the right framework to consider in the overcomplete regime as these might not be learnable with polynomial samples even though they are identifiable. This further motivates the need for understanding HMMs with structured transition matrices. We provide a proof of Theorem 1 with more explicitly stated conditions in Appendix D.

For our positive results we focus on understanding properties of structured transition matrices which make learning tractable. To disentangle additional complications due to the choice of the observation matrix, we will assume that the observation matrix is drawn at random throughout the paper. Long-standing open problems on learning *aliased* HMMs (HMMs where multiple hidden states have identical output distributions) [7, 15, 23] hint that understanding learnability with respect to properties of the observation matrix is a daunting task in itself, and is perhaps best studied separately from understanding how properties of the transition matrix affect learning.

Our positive result on learnability (Theorem 2) depends on two natural graph-theoretic properties of the transition matrix. We consider transition matrices which are i) sparse (hidden states have constant degree) and ii) have small probability mass on cycles shorter than $10 \log_m n$ states—and show that these HMMs can be learned efficiently using tensor decomposition and the method of moments, given random observation matrices. The condition prohibiting short cycles might seem mysterious. Intuitively, we need this condition to ensure that the Markov Chain visits a sufficient large portion of the state space in a short interval of time, and in fact the condition stems from information-theoretic considerations. We discuss these further in Sections 2.4 and 3.1. We also discuss how our results relate to learning HMMs which capture long-term dependencies in their outputs, and introduce a new notion of how well an HMM captures long-term dependencies. These are discussed in Section 5.

We also show new identifiability results for sparse HMMs. These results provide a finer picture of identifiability than Huang et al. [14], as ours hold for sparse transition matrices which are not generic.

**Technical contribution.** To prove Theorem 2 we show that the Khatri-Rao product of dependent random vectors is well-conditioned under certain conditions. Previously, Bhaskara et al. [6] showed that the Khatri-Rao product of independent random vectors is well-conditioned to perform a smoothed analysis of tensor decomposition, their techniques however do not extend to the dependent case. For the dependent case, we show a similar result using a novel Markov chain coupling based argument which relates the condition number to the best coupling of output distributions of two random walks with disjoint starting distributions. The technique is outlined in Section 2.2.

**Related work.** Spectral methods for learning HMMs have been studied in Anandkumar et al. [3], Bhaskara et al. [5], Allman et al. [1], Hsu et al. [13], but these results require $m \geq n$. In Allman et al. [1], the authors show that that HMMs are identifiable given moments of continuous observations over a time interval of length $N = 2\tau + 1$ for some $\tau$ such that $\binom{\tau+m-1}{m-1} \geq n$. When $m \ll n$ this requires $\tau = \mathcal{O}(n^{1/m})$. Bhaskara et al. [5] give another bound on window size which requires $\tau = \mathcal{O}(n/m)$. However, with a output alphabet of size $m$, specifying all moments in a $N$ length continuous time interval requires $m^N$ time and samples, and therefore all of these approaches lead to exponential runtimes when $m$ is constant with respect to $n$. Also relevant is the work by Anandkumar et al. [4] on guarantees for learning certain latent variable models such as Gaussian mixtures in the overcomplete setting through tensor decomposition. As mentioned earlier, the work closest to ours is Huang et al. [14] who showed that generic HMMs are identifiable with $\tau = \mathcal{O}(\log_m n)$, which gives the first polynomial runtimes for the case when $m$ is constant.

**Outline.** Section 2 introduces the notation and setup. It also provides examples and a high-level overview of our proof approach. Section 3 states the learnability result, discusses our assumptions and HMMs which satisfy these assumptions. Section 4 contains our identifiability results for sparse HMMs. Section 5 discusses natural measures of long-term dependencies in HMMs. We conclude in Section 6. Proof details are deferred to the Appendix.

## 2 Setup and preliminaries

In this section we first introduce the required notation, and then outline the method of moments approach for parameter recovery. We also go over some examples to provide a better understanding of the classes of HMMs we aim to learn, and give a high level proof strategy.

### 2.1 Notation and preliminaries

We will denote the output at time $t$ by $y_t$ and the hidden state at time $t$ by $h_t$. Let the number of hidden states be $n$ and the number of observations be $m$. Assume that the output alphabet is $\{0, \ldots, m-1\}$ without loss of generality. Let $T$ be the transition matrix and $O$ be the observation matrix of the HMM, both of these are defined so that the columns add up to one. For any matrix $A$, we refer to the $i$th column of $A$ as $A_i$. $T'$ is defined as the transition matrix of the time-reversed Markov chain, but we do not assume reversibility and hence $T$ may not equal $T'$. Let $y_i^j = y_i, \ldots, y_j$ denote the sequence of outputs from time $i$ to time $j$. Let $l_i^j = l_i, \ldots, l_j$ refer to a string of length $i + j - 1$ over the output alphabet, denoting a particular output sequence from time $i$ to $j$. Define a bijective mapping $L$ which maps an output sequence $l_1^\tau \in \{0, \ldots, m-1\}^\tau$ into an index $L(l_1^\tau) \in \{1, \ldots, m^\tau\}$ and the associated inverse mapping $L^{-1}$.

Throughout the paper, we assume that the transition matrix $T$ is ergodic, and hence has a stationary distribution. We also assume that every hidden state has stationary probability at least $1/\text{poly}(n)$. This is a necessary condition, as otherwise we might not even visit all states in $\text{poly}(n)$ samples. We also assume that the output process of the HMM is stationary. A stochastic process is stationary if the distribution of any subset of random variables is invariant with respect to shifts in the time index—that is, $\mathbb{P}[y_{-\tau}^\tau = l_{-\tau}^\tau] = \mathbb{P}[y_{-\tau+T}^{\tau+T} = l_{-\tau}^\tau]$ for any $\tau, T$ and string $l_{-\tau}^\tau$. This is true if the initial hidden state is chosen according to the stationary distribution.

Our results depend on the conditioning of the matrix $T$ with respect to the $\ell_1$ norm. We define $\sigma_{\min}^{(1)}(T)$ as the minimum $\ell_1$ gain of the transition matrix $T$ over all vectors $x$ having unit $\ell_1$ norm (not just non-negative vectors $x$, for which the ratio would always be 1):

$$\sigma_{\min}^{(1)}(T) = \min_{x \in \mathbb{R}^n} \frac{\|Tx\|_1}{\|x\|_1}$$

$\sigma_{\min}^{(1)}(T)$ is also a natural parameter to measure the long-term dependence of the HMM—if $\sigma_{\min}^{(1)}(T)$ is large then $T$ preserves significant information about the distribution of hidden states at time 0 at a future time $t$, for all initial distributions at time 0. We discuss this further in Section 5.

### 2.2 Method of moments for learning HMMs

Our algorithm for learning HMMs follows the method of moments based approach, outlined for example in Anandkumar et al. [2] and Huang et al. [14]. In contrast to the more popular Expectation-Maximization (EM) approach which can suffer from slow convergence and local optima [21], the method of moments approach ensures guaranteed recovery of the parameters under mild conditions. More details about tensor decomposition and the method of moments approach to learning HMMs can be found in Appendix A.

The method of moments approach to learning HMMs has two high-level steps. In the first step, we write down a tensor of empirical moments of the data, such that the factors of the tensor correspond to parameters of the underlying model. In the second step, we perform tensor decomposition to recover the factors of the tensor—and then recover the parameters of the model from the factors. The key fact that enables the second step is that tensors have a unique decomposition under mild conditions on their factors, for example tensors have a unique decomposition if all the factors are full rank. The uniqueness of tensor decomposition permits unique recovery of the parameters of the model.

We will learn the HMM using the moments of observation sequences $y_{-\tau}^\tau$ from time $-\tau$ to $\tau$. Since the output process is assumed to be stationary, the distribution of outputs is the same for

any contiguous time interval of the same length, and we use the interval $-\tau$ to $\tau$ in our setup for convenience. We call the length of the observation sequences used for learning the *window length* $N = 2\tau + 1$. Since the number of samples required to estimate moments over a window of length $N$ is $m^N$, it is desirable to keep $N$ small. Note that to ensure polynomial runtime and sample complexity for the method of moments approach, the window length $N$ must be $\mathcal{O}(\log_m n)$.

We will now define our moment tensor. Given moments over a window of length $N = 2\tau + 1$, we can construct the third-order moment tensor $M \in \mathbb{R}^{m^\tau \times m^\tau \times m}$ using the mapping $L$ from strings of outputs to indices in the tensor:

$$M_{(L(l_1^\tau), L(l_{-1}^{-\tau}), l_0)} = \mathbb{P}[y_{-\tau}^\tau = l_{-\tau}^\tau].$$

$M$ is simply the tensor of the moments of the HMM over a window length $N$, and can be estimated directly from data. We can write $M$ as an outer product because of the Markov property:

$$M = A \otimes B \otimes C$$

where $A \in \mathbb{R}^{m^\tau \times n}, B \in \mathbb{R}^{m^\tau \times n}, C \in \mathbb{R}^{m \times n}$ are defined as follows (here $h_0$ denotes the hidden state at time 0):

$$A_{L(l_1^\tau), i} = \mathbb{P}[y_1^\tau = l_1^\tau \mid h_0 = i]$$
$$B_{L(l_{-1}^{-\tau}), i} = \mathbb{P}[y_{-1}^{-\tau} = l_{-1}^{-\tau} \mid h_0 = i]$$
$$C_{l_0, i} = \mathbb{P}[y_0 = l, h_0 = i]$$

$T$ and $O$ can be related in a simple manner to $A$, $B$ and $C$. If we can decompose the tensor $M$ into the factors $A$, $B$ and $C$, we can recover $T$ and $O$ from $A$, $B$ and $C$. Kruskal's condition [18] guarantees that tensors have a unique decomposition whenever $A$ and $B$ are full rank and no two column of $C$ are the same. We refer the reader to Appendix A for more details, specifically Algorithm 1.

## 2.3 High-level proof strategy

As the transition and observation matrices can be recovered from the factors of the tensors, our goal is to analyze the conditions under which the tensor decomposition step works provably. Note that the factor matrix $A$ is the likelihood of observing each sequence of observations conditioned on starting at a given hidden state. We'll refer to $A$ as the *likelihood matrix* for this reason. $B$ is the equivalent matrix for the time-reversed Markov chain. If we show that $A$, $B$ are full rank and no two columns of $C$ are the same, then the HMM can be learned provided the exact moments using the simultaneous diagonalization algorithm, also known as Jennrich's algorithm (see Algorithm 1). We show this property for our identifiability results. For our learnability results, we show that the matrices $A$ and $B$ are well-conditioned (have condition numbers polynomial in $n$), which implies learnability from polynomial samples. This is the main technical contribution of the paper, and requires analyzing the condition number of the Khatri-Rao product of dependent random vectors. Before sketching the argument, we first introduce some notation. We can define $A^{(t)}$ as the likelihood matrix over $t$ steps:

$$A^{(t)}_{L(l_1^t), i} = \mathbb{P}[y_1^t = l_1^t \mid h_0 = i].$$

$A^{(t)}$ can be recursively written down as follows:

$$A^{(0)} = OT, \ A^{(t)} = (O \odot A^{(t-1)})T \tag{1}$$

where $A \odot B$, denotes the Khatri-Rao product of the matrices $A$ and $B$. If $A$ and $B$ are two matrices of size $m_1 \times r$ and $m_2 \times r$ then the Khatri-Rao product is a $m_1 m_2 \times r$ matrix whose $i$th column is the outer product $A_i \otimes B_i$ flattened into a vector. Note that $A^{(\tau)}$ is the same as $A$. We now sketch our argument for showing that $A^{(\tau)}$ is well-conditioned under appropriate conditions.

**Coupling random walks to analyze the Khatri-Rao product.** As mentioned in the introduction, in this paper we are interested in the setting where the transition matrix is fixed but the observation matrix is drawn at random. If we could draw fresh random matrices $O$ at each time step of the recursion in Eq. 1, then $A$ would be well-conditioned by the smoothed analysis of the Khatri-Rao product due to Bhaskara et al. [6]. However, our setting is significantly more difficult, as we do not have access to fresh randomness at each time step, so the techniques of Bhaskara et al. [6] cannot be applied here. As pointed out earlier, the condition number of $A$ in this scenario depends crucially on the transition matrix $T$, as $A$ is not even full rank if $T = I$.

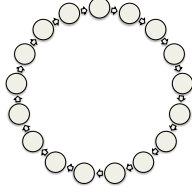

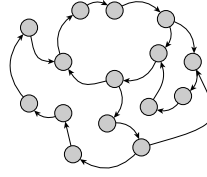

(a) Transition matrix is a cycle, or a permutation on the hidden states.

(b) Transition matrix is a random walk on a graph with small degree and no short cycles.

Figure 1: Examples of transition matrices which we can learn, refer to Section 2.4 and Section 3.2.

Instead, we analyze $A$ by a coupling argument. To get some intuition for this, note that if $A$ does not have full rank, then there are two disjoint sets of columns of $A$ whose linear combinations are equal, and these combination weights can be used to setup the initial states of two random walks defined by the transition matrix $T$ which have the same output distribution for $\tau$ time steps. More generally, if $A$ is ill-conditioned then there are two random walks with disjoint starting states which have very similar output distributions. We show that if two random walks have very similar output distributions over $\tau$ time steps for a randomly chosen observation matrix $O$, then most of the probability mass in these random walks can be coupled. On the other hand, if $(\sigma_{\min}^{(1)}(T))^{\tau}$ is sufficiently large, the total variational distance between random walks starting at two different starting states must be at least $(\sigma_{\min}^{(1)}(T))^{\tau}$ after $\tau$ time steps, and so there cannot be a good coupling, and $A$ is well-conditioned. We provide a sketch of the argument for a simple case in Appendix 1 before we prove Theorem 2.

### 2.4 Illustrative examples

We now provide a few simple examples which will illustrate some classes of HMMs we can and cannot learn. We first provide an example of a class of simple HMMs which can be handled by our results, but has non-generic transition matrices and hence does not fit into the framework of Huang et al. [14]. Consider an HMM where the transition matrix is a permutation or cyclic shift on the hidden states (see Fig. 1a). Our results imply that such HMMs are learnable in polynomial time from polynomial samples if the output distributions of the hidden states are chosen at random. We will try to provide some intuition about why an HMM with the transition matrix as in Fig. 1a should be efficiently learnable. Let us consider the simple case when the the outputs are binary (so $m = 2$) and each hidden state deterministically outputs a 0 or a 1, and is labeled by a 0 or a 1 accordingly. If the labels are assigned at random, then with high probability the string of labels of any continuous sequence of $2\log_2 n$ hidden states in the cycle in Fig. 1a will be unique. This means that the output distribution in a $2\log_2 n$ time window is unique for every initial hidden state, and it can be shown that this ensures that the moment tensor has a unique factorization. By showing that the output distribution in a $2\log_2 n$ time window is very different for different initial hidden states—in addition to being unique—we can show that the factors of the moment tensor are well-conditioned, which allows recovery with efficient sample complexity. As another slightly more complex example of an HMM we can learn, Fig. 1b depicts an HMM whose transition matrix is a random walk on a graph with small degree and no short cycles. Our learnability result can handle such HMMs having structured transition matrices.

As an example of an HMM which cannot be learned in our framework, consider an HMM with transition matrix $T = I$ and binary observations ($m = 2$), see Fig. 2a. In this case, the probability of an output sequence only depends on the total number of zeros or ones in the sequence. Therefore, we only get $t$ independent measurements from windows of length $t$, hence windows of length $\mathcal{O}(n)$ instead of $\mathcal{O}(\log_2 n)$ are necessary for identifiability (also refer to Blischke [8] for more discussions on this case). More generally, we prove in Proposition 1 that for small $m$ a transition matrix composed only of cycles of constant length (see Fig. 2b) requires the window length to be polynomial in $n$ to become identifiable.

**Proposition 1.** *Consider an HMM on $n$ hidden states and $m$ observations with the transition matrix being a permutation composed of cycles of length $c$. Then windows of length $O(n^{1/m^c})$ are necessary for the model to be identifiable, which is polynomial in $n$ for constant $c$ and $m$.*

The root cause of the difficulty in learning HMMs having short cycles is that they do not visit a large enough portion of the state space in $\mathcal{O}(\log_m n)$ steps, and hence moments over a $\mathcal{O}(\log_m n)$ time

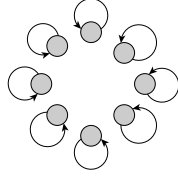 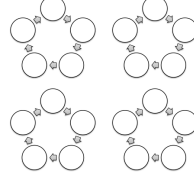

(a) Transition matrix is the identity on 8 hidden states.

(b) Transition matrix is a union of 4 cycles, each on 5 hidden states.

Figure 2: Examples of transition matrices which do not fit in our framework. Proposition 1 shows that such HMMs where the transition matrix is composed of a union of cycles of constant length are not even identifiable from short windows of length $\mathcal{O}(\log_m n)$

window do not carry sufficient information for learning. Our results cannot handle such classes of transition matrices, also see Section 3.1 for more discussion.

# 3 Learnability results for overcomplete HMMs

In this section, we state our learnability result, discuss the assumptions and provide examples of HMMs which satisfy these assumptions. Our learnability results hold under the following conditions:

**Assumptions:** For fixed constants $c_1, c_2, c_3 > 1$, the HMM satisfies the following properties for some $c > 0$:

1. *Transition matrix is well-conditioned:* Both $T$ and the transition matrix $T'$ of the time reversed Markov Chain are well-conditioned in the $\ell_1$-norm: $\sigma_{\min}^{(1)}(T), \sigma_{\min}^{(1)}(T') \geq 1/m^{c/c_1}$

2. *Transition matrix does not have short cycles:* For both $T$ and $T'$, every state visits at least $10\log_m n$ states in $15\log_m n$ time except with probability $\delta_1 \leq 1/n^c$.

3. *All hidden states have small "degree":* There exists $\delta_2$ such that for every hidden state $i$, the transition distributions $T_i$ and $T_i'$ have cumulative mass at most $\delta_2$ on all but $d$ states, with $d \leq m^{1/c_2}$ and $\delta_2 \leq 1/n^c$. Hence this is a soft "degree" requirement.

4. *Output distributions are random and have small support :* There exists $\delta_3$ such that for every hidden state $i$ the output distribution $O_i$ has cumulative mass at most $\delta_3$ on all but $k$ outputs, with $k \leq m^{1/c_3}$ and $\delta_3 \leq 1/n^c$. Also, the output distribution $O_i$ is drawn uniformly on these $k$ outputs.

The constants $c_1, c_2, c_3$ are can be made explicit, for example, $c_1 = 20, c_2 = 16$ and $c_3 = 10$ works. Under these conditions, we show that HMMs can be learned using polynomially many samples:

**Theorem 2.** *If an HMM satisfies the above conditions, then with high probability over the choice of O, the parameters of the HMM are learnable to within additive error $\epsilon$ with observations over windows of length $2\tau + 1, \tau = 15\log_m n$, with the sample complexity poly$(n, 1/\epsilon)$.*

Appendix C also states a corollary of Theorem 2 in terms of the minimum singular value $\sigma_{\min}(T)$ of the matrix $T$, instead of $\sigma_{\min}^{(1)}(T)$. We discuss the conditions for Theorem 2 next, and subsequently provide examples of HMMs which satisfy these conditions.

## 3.1 Discussion of the assumptions

1. *Transition matrix is well-conditioned:* Note that singular transition matrices might not even be identifiable. Moreover, Mossel and Roch [20] showed that learning HMMs with singular transition matrices is as hard as learning parity with noise, which is widely conjectured to be computationally hard. Hence, it is necessary to exclude at least some classes of ill-conditioned transition matrices.

2. *Transition matrix does not have short cycles:* Due to Proposition 1, we know that a HMM might not even be identifiable from short windows if it is composed of a union of short cycles, hence we expect a similar condition for learning the HMM with polynomial samples; though there is a gap between the upper and lower bounds in terms of the probability mass which is allowed on the short cycles. We performed some simulations to understand how the length of cycles in the transition matrix and the probability mass assigned to short cycles affects the condition number of the likelihood matrix $A$; recall that the condition number of $A$ determines the stability of the method of moments

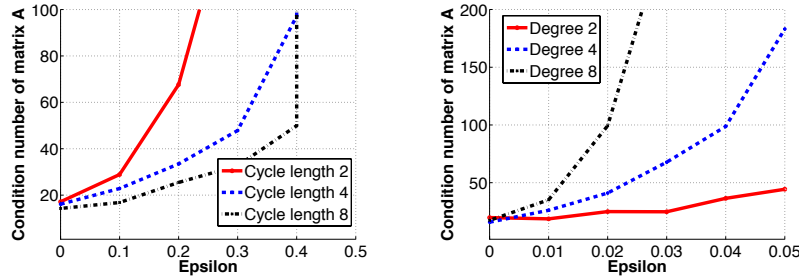

(a) The conditioning becomes worse when cycles are smaller or when more probability mass $\epsilon$ is put on short cycles.

(b) The conditioning becomes worse as the degree increases, and when more probabiltiy mass $\epsilon$ is put on the dense part of $T$.

Figure 3: Experiments to study the effect of sparsity and short cycles on the learnability of HMMs. The condition number of the likelihood matrix $A$ determines the stability or sample complexity of the method of moments approach. The condition numbers are averaged over 10 trials.

approach. We take the number of hidden states $n = 128$, and let $P_{128}$ be a cycle on the $n$ hidden states (as in Fig. 1a). Let $P_c$ be a union of short cycles of length $c$ on the $n$ states (refer to Fig. 2b for an example). We take the transition matrix to be $T = \epsilon P_c + (1 - \epsilon)P_{128}$ for different values of $c$ and $\epsilon$. Fig. 3a shows that the condition number of $A$ becomes worse and hence learning requires more samples if the cycles are shorter in length, and if more probability mass is assigned to the short cycles, hinting that our conditions are perhaps not be too stringent.

3. *All hidden states have a small degree:* Condition 3 in Theorem 2 can be reinterpreted as saying that the transition probabilities out of any hidden state must have mass at most $1/n^{1+c}$ on any hidden state except a set of $d$ hidden states, for any $c > 0$. While this soft constraint is weaker than a hard constraint on the degree, it natural to ask whether any sparsity is necessary to learn HMMs. As above, we carry out simulations to understand how the degree affects the condition number of the likelihood matrix $A$. We consider transition matrices on $n = 128$ hidden states which are a combination of a dense part and a cycle. Define $P_{128}$ to be a cycle as before. Define $G_d$ as the adjacency matrix of a directed regular graph with degree $d$. We take the transition matrix $T = \epsilon G_d + (1 - \epsilon d)P_{128}$. Hence the transition distribution of every hidden state has mass $\epsilon$ on a set of $d$ neighbors, and the residual probability mass is assigned to the permutation $P_{128}$. Fig. 3b shows that the condition number of $A$ becomes worse as the degree $d$ becomes larger, and as more probability mass $\epsilon$ is assigned to the dense part $G_d$ of the transition matrix $T$, providing some weak evidence for the necessity of Condition 3. Also, recall that Theorem 1 shows that HMMs where the transition matrix is a random walk on an undirected regular graph with large degree (degree polynomial in $n$) cannot be learned using polynomially many samples if $m$ is constant with respect to $n$. However, such graphs have all eigenvalues except the first one to be less than $O(1/\sqrt{d})$, hence it is not clear if the hardness of learning depends on the large degree itself or is only due to $T$ being ill-conditioned. More concretely, we pose the following open question:

**Open question:** Consider an HMM with a transition matrix $T = (1 - \epsilon)P + \epsilon U$, where $P$ is the cyclic permutation on $n$ hidden states (such as in Fig. 1a) and $U$ is a random walk on a undirected, regular graph with large degree (polynomial in $n$) and $\epsilon > 0$ is a constant. Can this HMM be learned using polynomial samples when $m$ is small (constant) with respect to $n$? This example approximately preserves $\sigma_{\min}(T)$ by the addition of the permutation, and hence the difficulty is only due to the transition matrix having large degree.

4. *Output distributions are random and have small support:* As discussed in the introduction, if we do not assume that the observation matrices are random, then even simple HMMs with a cycle or permutation as the transition matrix might require long windows even to become identifiable, see Fig. 4. Hence some assumptions on the output distribution do seem necessary for learning the model from short time windows, though our assumptions are probably not tight. For instance, the assumption that the output distributions have a small support makes learning easier as it leads to the outputs being more discriminative of the hidden states, but it is not clear that this is a necessary assumption. Ideally, we would like to prove our learnability results under a *smoothed* model for $O$, where an adversary is allowed to see the transition matrix $T$ and pick any worst-case $O$, but random noise is then added to

the output distributions, which limits the power of the adversary. We believe our results should hold under such a smoothed setting, but set this aside for future work.

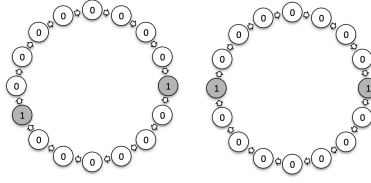

Figure 4: Consider two HMMs with transition matrices being cycles on $n = 16$ states with binary outputs, and outputs conditioned on the hidden states are deterministic. The states labeled as 0 always emit a 0 and the states labeled as 1 always emit a 1. The two HMMs are not distinguishable from windows of length less than 8. Hence with worst case $O$ even simple HMMs like the cycle could require long windows to even become identifiable.

### 3.2 Examples of transition matrices which satisfy our assumptions

We revisit the examples from Fig. 1a and Fig. 1b, showing that they satisfy our assumptions.

1. *Transition matrices where the Markov Chain is a permutation:* If the Markov chain is a permutation with all cycles longer than $10 \log_m n$ then the transition matrix obeys all the conditions for Theorem 2. This is because all the singular values of a permutation are 1, the degree is 1 and all hidden states visit $10 \log_m n$ different states in $15 \log_m n$ time steps.

2. *Transition matrices which are random walks on graphs with small degree and large girth:* For directed graphs, Condition 2 can be equivalently stated as that the graph representation of the transition matrix has a large girth (girth of a graph is defined as the length of its shortest cycle).

3. *Transition matrices of factorial HMMs:* Factorial HMMs [12] factor the latent state at any time into $D$ dimensions, each of which independently evolves according to a Markov process. For $D = 2$, this is equivalent to saying that the hidden states are indexed by two labels $(i, j)$ and if $T_1$ and $T_2$ represent the transition matrices for the two dimensions, then $\mathbb{P}[(i_1, j_1) \to (i_2, j_2)] = T_1(i_2, i_1) T_2(j_2, j_1)$. This naturally models settings where there are multiple latent concepts which evolve independently. The following properties are easy to show:

   1. If either of $T_1$ or $T_2$ visit $N$ different states in $15 \log_m n$ time steps with probability $(1 - \delta)$, then $T$ visits $N$ different states in $15 \log_m n$ time steps with probability $(1 - \delta)$.
   2. $\sigma_{\min}(T) = \sigma_{\min}(T_1)\sigma_{\min}(T_2)$
   3. If all hidden states in $T_1$ and $T_2$ have mass at most $\delta$ on all but $d_1$ states and $d_2$ states respectively, then $T$ has mass at most $2\delta$ on all but $d_1 d_2$ states.

Therefore, factorial HMMs are learnable with random $O$ if the underlying processes obey conditions similar to the assumptions for Theorem 2. If both $T_1$ and $T_2$ are well-conditioned and at least one of them does not have short cycles, and either has small degree, then $T$ is learnable with random $O$.

## 4 Identifiability of HMMs from short windows

As it is not obvious that some of the requirements for Theorem 2 are necessary, it is natural to attempt to derive stronger results for just identifiability of HMMs having structured transition matrices. In this section, we state our results for identifiability of HMMs from windows of size $\mathcal{O}(\log_m n)$. Huang et al. [14] showed that all HMMs except those belonging to a measure zero set become identifiable from windows of length $2\tau + 1$ with $\tau = 8\lceil \log_m n \rceil$. However, the measure zero set itself might possibly contain interesting classes of HMMs (see Fig. 1), for example sparse HMMs also belong to a measure zero set. We refine the identifiability results in this section, and show that a natural sparsity condition on the transition matrix guarantees identifiability from short windows. Given any transition matrix $T$, we regard $T$ as being supported by a set of indices $\mathcal{S}$ if the non-zero entries of $T$ all lie in $\mathcal{S}$. We now state our result for identifiability of sparse HMMs.

**Theorem 3.** *Let $\mathcal{S}$ be a set of indices which supports a permutation where all cycles have at least $2\lceil \log_m n \rceil$ hidden states. Then the set $\mathcal{T}$ of all transition matrices with support $\mathcal{S}$ is identifiable from windows of length $4\lceil \log_m n \rceil + 1$ for all observation matrices $O$ except for a measure zero set of transition matrices in $\mathcal{T}$ and observation matrices $O$.*

We hypothesize that excluding a measure zero set of transition matrices in Theorem 3 should not be necessary as long as the transition matrix is full rank, but are unable to show this. Note that our result on identifiability is more flexible in allowing short cycles in transition matrices than Theorem 2, and is closer to the lower bound on identifiability in Proposition 1.

We also strengthen the result of Huang et al. [14] for identifiability of generic HMMs. Huang et al. [14] conjectured that windows of length $2\lceil \log_m n \rceil + 1$ are sufficient for generic HMMs to be identifiable. The constant 2 is the information theoretic bound as an HMM on $n$ hidden states and $m$ outputs has $\mathcal{O}(n^2 + nm)$ independent parameters, and hence needs observations over a window of size $2\lceil \log_m n \rceil + 1$ to be uniquely identifiable. Proposition 2 settles this conjecture, proving the optimal window length requirement for generic HMMs to be identifiable. As the number of possible outputs over a window of length $t$ is $m^t$, the size of the moment tensor in Section 2.2 is itself exponential in the window length. Therefore even a factor of 2 improvement in the window length requirement leads to a quadratic improvement in the sample and time complexity.

**Proposition 2.** *The set of all HMMs is identifiable from observations over windows of length* $2\lceil \log_m n \rceil + 1$ *except for a measure zero set of transition matrices $T$ and observation matrices $O$.*

## 5   Discussion on long-term dependencies in HMMs

In this section, we discuss long-term dependencies in HMMs, and show how our results on overcomplete HMMs improve the understanding of how HMMs can capture long-term dependencies, both with respect to the Markov chain and the outputs. Recall the definition of $\sigma_{\min}^{(1)}(T)$:

$$\sigma_{\min}^{(1)}(T) = \min_{x \in \mathbb{R}^n} \frac{\|Tx\|_1}{\|x\|_1}$$

We claim that if $\sigma_{\min}^{(1)}(T)$ is large, then the transition matrix preserves significant information about the distribution of hidden states at time 0 at a future time $t$, for all initial distributions at time 0. Consider any two distributions $p_0$ and $q_0$ at time 0. Let $p_t$ and $q_t$ be the distributions of the hidden states at time $t$ given that the distribution at time 0 is $p_0$ and $q_0$ respectively. Then the $\ell_1$ distance between $p_t$ and $q_t$ is $\|p_t - q_t\|_1 \geq (\sigma_{\min}^{(1)}(T))^t \|p_0 - q_0\|_1$, verifying our claim. It is interesting to compare this notion with the mixing time of the transition matrix. Defining mixing time as the time until the $\ell_1$ distance between any two starting distributions is at most $1/2$, it follows that the mixing time $\tau_{\mathrm{mix}} \geq 1/\log(1/\sigma_{\min}^{(1)}(T))$, therefore if $\sigma_{\min}^{(1)}(T))$ is large then the chain is slowly mixing. However, the converse is not true—$\sigma_{\min}^{(1)}(T)$ might be small even if the chain never mixes, for example if the graph is disconnected but the connected components mix very quickly. Therefore, $\sigma_{\min}^{(1)}(T)$ is possibly a better notion of the long-term dependence of the transition matrix, as it requires that information is preserved about the past state "in all directions".

Another reasonable notion of the long-term dependence of the HMM is the long-term dependence in the output process instead of in the hidden Markov chain, which is the utility of past observations when making predictions about the distant future (given outputs $y_{-\infty}, \ldots, y_1, y_2, \ldots, y_t$, at time $t$ how far back do we need to remember about the past to make a good prediction about $y_t$?). This does not depend in a simple way on the $T$ and $O$ matrices, but we do note that if the Markov chain is fast mixing then the output process can certainly not have long-term dependencies. We also note that with respect to long-term dependencies in the output process, the setting $m \ll n$ seems to be much more interesting than when $m$ is comparable to $n$. The reason is that in the small output alphabet setting we only receive a small amount of information about the true hidden state at each step, and hence longer windows are necessary to infer the hidden state and make a good prediction. We also refer the reader to Kakade et al. [16] for related discussions on the memory of output processes of HMMs.

## 6   Conclusion and Future Work

The setting where the output alphabet $m$ is much smaller than the number of hidden states $n$ is well-motivated in practice and seems to have several interesting theoretical questions about new lower bounds and algorithms. Though some of our results are obtained in more restrictive conditions than seems necessary, we hope the ideas and techniques pave the way for much sharper results in this setting. Some open problems which we think might be particularly useful for improving our understanding is relaxing the condition on the observation matrix being random to some structural constraint on the observation matrix (such as on its Kruskal rank), and more thoroughly investigating the requirement for the transition matrix being sparse and not having short cycles.

## Footnotes

[1]Parity with noise is information theoretically easy given observations over a window of length at least the number of inputs to the parity. This is linear in the number of hidden states of the parity with noise HMM, whereas Theorem 1 says that the sample complexity must be super polynomial for any polynomial sized window.

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
