[Supplementary Material 1]

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

## A Discussion on tensor decomposition and method of moments for learning HMMs

We begin with some notation and basic facts about tensor decomposition in Section A.1. We will follow by outlining the well-known tensor decomposition based approach to HMM learning in Section A.2.

### A.1 Tensor preliminaries

Given a 3rd order rank $k$ tensor $M \in \mathbb{R}^{d_1 \times d_2 \times d_3}$, it can be written in terms of its factor matrices $A, B$ and $C$:

$$M = \sum_{i \in [k]} A_i \otimes B_i \otimes C_i$$

where $A_i$ denotes the $i$th column of a matrix $A$. Here $\otimes$ denotes the tensor product: if $a, b, c \in \mathbb{R}^d$ then $a \otimes b \otimes c \in \mathbb{R}^{d \times d \times d}$ and $(a \otimes b \otimes c)_{ijk} = a_i b_j c_k$. We refer to different dimensions of a tensor as the *modes* of the tensor.

We denote $M_{(k)}$ as the mode $k$ matricization of the tensor, which is the flattening of the tensor along the $k$th direction obtained by stacking all the matrix slices together. For example $T_{(1)}$ denotes flattening of a tensor $T \in \mathbb{R}^{n \times m \times p}$ to a $(n \times mp)$ matrix. Recall that we denote the Khatri-Rao product of two matrices $A$ and $B$ as $(A \odot B)_i = (A_i \otimes B_i)_{(1)}$, where $(A_i \otimes B_i)_{(1)}$ denotes the flattening of the matrix $A_i \otimes B_i$ into a row vector. We denote the set $\{1, 2, \cdots, k\} = [k]$.

Kruskal's condition [18] says that if $A$ and $B$ are full rank and no two rows of $C$ are linearly dependent, then $M$ can be efficiently decomposed into the factors $A, B, C$ and the decomposition is unique upto scaling and permutation. The simultaneous decomposition algorithm [9, 19] (Algorithm 1), is a well known algorithm to decompose tensors which satisfy Kruskal's condition.

### A.2 Learning HMMs using tensor decomposition

The transition matrix $T \in \mathbb{R}^{n \times n}$ for the HMM is defined as follows-
$$\mathbb{P}[h_{t+1} = i | h_t = j] = T_{i,j} \ \forall \ i, j$$
The observation matrix $O \in \mathbb{R}^{m \times n}$ is the probability of observing each output conditioned on the hidden state-
$$\mathbb{P}[y_t = i | h_t = j] = O_{i,j} \ \forall \ i, j$$
Also, given the transition matrix $T$, we define $T'$ as the time reversed Markov Chain. If the Markov Chain is time-reversible then $T = T'$.

Our framework for HMM learning via tensor decomposition is identical to Huang et al. [14]. Algorithm 1 describes how to compute $T$ and $O$ from the factors $A, B$ and $C$, we refer the reader to Huang et al. [14] for more details.

## B Additional proofs for Section 4: Identifiability results

**Theorem 3.** *Let $\mathcal{S}$ be a set of indices which supports a permutation where all cycles have at least $2\lceil \log_m n \rceil$ hidden states. Then the set $\mathcal{T}$ of all transition matrices with support $\mathcal{S}$ is identifiable from windows of length $4\lceil \log_m n \rceil + 1$ for all observation matrices $O$ except for a measure zero set of transition matrices in $\mathcal{T}$ and observation matrices $O$.*

*Proof sketch.* Recall from Section 2.2 that the main task is to show that the likelihood matrix $A$ is full rank. The proof uses basic algebraic geometry, and the main idea used is analogous to the following fact about polynomials: either a polynomial is a zero polynomial or it has finitely many roots which will lie in a measure zero set. The determinant of the likelihood matrix $A$ (or of sub-matrices of $A$ if $A$ is rectangular) is a polynomial in the entries of $T$ and $O$, hence we only need to show that the polynomial is not a zero polynomial. To show that a polynomial is not a zero polynomial, it is sufficient to find one instance of the variables which makes the polynomial non-zero. Hence we only need to find some particular $T$ and $O$ such that the determinant is not 0. We find such a $T$ and $O$ using the fact that $\mathcal{S}$ supports a permutation which does not have short cycles.

---

**Algorithm 1** Learning HMMs with $m \ll n$ ([14])

---

**Input:** Moment tensor $M \in \mathbb{R}^{m^\tau \times m^\tau \times m}$ over a window of length $\tau$
**Output:** Estimates $\hat{T}$ and $\hat{O}$

**Tensor decomposition using simultaneous diagonalization:**

1. Choose $a, b \in \mathbb{R}^d$ uniformly at random. Project $M$ along the 3rd dimension to obtain $X, Y$ with $X_{i,j} = \sum_k M_{i,j,k} a_k$ and $Y_{i,j} = \sum_k M_{i,j,k} b_k$.

2. Compute the eigendecomposition of $X(Y)^{-1}$ and $Y(X)^{-1}$. Let the columns of $A$ and $B$ to be the eigenvectors of $X(Y)^{-1}$ and $Y(X)^{-1}$ respectively. Pair them corresponding to reciprocal eigenvalues, and scale $A$ and $B$ to be column-stochastic.

3. Let $M_{(3)} \in \mathbb{R}^{d^{2\tau} \times d}$ be the mode 3 matricization of $M$. Set $C = M_{(3)}((A \odot B)^\dagger)^T$.

**Estimating $T$ and $O$ from tensor factors:**

1. Estimate $O$ by normalizing $C$ to be stochastic, i.e. $\hat{O}_{[:,i]} = C_{:,i}/(e^T C_{[:,i]})$ for all $i$.

2. Marginalize $A$ over the final time step to obtain $A^{(\tau-1)}$.

3. Estimate $T = (O \odot A^{(\tau-1)})^\dagger A$,

---

*Proof.* We first show that the matrices $A$, $B$ and $A' = (O \odot A^{(\tau-1)})$ become full rank using observations over a window of length $N = 2\tau + 1$, where $\tau = 2\lceil \log_m n \rceil$.

Let's fix the window length to be $N = 2\tau + 1$, where $\tau = 2\lceil \log_m n \rceil$. We choose the transition matrix $T$ to be the permutation which is supported by $\mathcal{S}$. By the requirement in Theorem 3, $T$ is composed of cycles all of which have at least $\tau$ hidden states. Without loss of generality, assume that all cycles in $T$ are composed of sequences of hidden states $\{h_i, h_{i+1}, \cdots, h_{j-1}, h_j\}$ for $i, j \in [n], i \leq j$. To further simplify notation we also assume, without loss of generality, that each cycle of $T$ is a cyclic shift of the hidden states $\{h_i, h_{i+1}, \cdots, h_{j-1}, h_j\}$.

To show that $A, B$ and $A'$ are full rank, it is sufficient to show that they become full rank for some particular choice of $O$. Say that we choose each hidden state $h_i$ to always deterministically output some character $a_i$. We show that $A$ is full-rank for some assignment of outputs to hidden states. The same argument will also imply that $B$ and $A'$ are full rank. Because the transition matrix is a permutation, Markov chains starting in two different starting states are at different hidden states at every time step. Also, because all cycles in the permutation are longer than $\tau$, at every time step at least one of the two states visited by the two initial starting states has not been visited so far by either of the two initial starting states. Hence, the probability of the emissions corresponding to the two initial starting states being the same at any time step is $(1/m)$. Therefore, the probability that the emissions for two different hidden states across a $\tau$ length windows is the same is $(1/m)^{\lceil 2 \log_m n \rceil} \leq 1/n^2$. As the total number of pairs of hidden states is $n(n-1)/2$, by a union bound, the probability there exists some observation matrix such that the emissions for the $\lceil 2 \log_m n \rceil$ window corresponding to all initial states are different is strictly great than 0. By choosing the $O$ which has this property, the matrix $A$ is full rank. By the same reasoning, $A'$ and $B$ are also full rank.

Hence, as we have shown that there exists some transition matrix $T$ which is supported by $\mathcal{S}$ and observation matrix $O$ such that $A, A'$ and $B$ become full rank. Hence $A, A'$ and $B$ are full rank except for a measure zero set of $T$ and $O$. As the matrix $C$ is linearly independence except for a measure zero set of $O$, Kruskal's condition is satisfied except for a measure zero set of $T$ and $O$, and $A'$ is full rank, hence due to Algorithm 1, the set of HMMs is identifiable except for a measure zero set of $T$ and $O$. This concludes the proof.

$\square$

**Proposition 2.** *The set of all HMMs is identifiable from observations over windows of length* $2\lceil \log_m n \rceil + 1$ *except for a measure zero set of transition matrices $T$ and observation matrices $O$.*

*Proof.* The proof idea is similar to Theorem 3. We will choose some particular choice of $T$ and $O$ such that the matrices $A, B$ and $A'$ become full rank. Consider $T$ to be the permutation on $n$ hidden states which performs a cyclic shift of the hidden states i.e. $T_{ij} = 1$ if $j = (i+1) \mod n$ for $0 \leq i, j \leq k-1$ and $T_{ij}$ is 0 otherwise. We show that for this particular choice of $T$, there exists a

choice of $O$ such that $A$ becomes full-rank for $t = \lceil \log_m n \rceil + 1$. As in the proof of Proposition 3, we choose each hidden state to deterministically output a character, hence each column of $O$ has one non-zero entry. We show that $A$ is full-rank for some choice of $O$, the same argument will also imply that $B$ and $A'$ are full rank for that $O$.

Because the transition matrix is a cyclic shift, we can reformulate the problem of finding a suitable $O$ as that of finding a length $n$ $m$-ary string all of whose length $\tau = \lceil \log_m n \rceil$ cyclic-substrings are unique (a substring is defined as a continuous subsequence), treating the string as cyclic so that the ends wrap around. De Bruijn sequences have exactly this property. A De Brujin sequence of length $k$ is a cyclic sequence in which every possible length $\lfloor \log_m k \rfloor$ $m$-ary string occurs exactly once as a substring. Furthermore, these sequences also have the property that all substrings longer than $\lfloor \log_m k \rfloor$ are unique [10]. Hence choosing a De Bruijn sequence of length $k = n$ ensures that all substrings of length $\lceil \log_m n \rceil \geq \lfloor \log_m n \rfloor$ are unique.

Hence we have shown that there exists a choice of $O$ such that the matrix $A$ becomes full rank with windows of length $2\lceil \log_m n \rceil + 1$ for some choice of $O$. Hence by the same argument as in the proof of Proposition 3, all HMMs except those belonging to a measure zero set of $T$ and $O$ become identifiable with windows of length $2\lceil \log_m n \rceil + 1$. $\qquad\square$

**Proposition 1.** *Consider an HMM on $n$ hidden states and $m$ observations with the transition matrix being a permutation composed of cycles of length $c$. Then windows of length $O(n^{1/m^c})$ are necessary for the model to be identifiable, which is polynomial in $n$ for constant $c$ and $m$.*

*Proof.* There are $m^c$ possible outputs over a window of length $c$. Define a larger alphabet of size $m^c$ which denotes output sequences over a window of length $c$. Dividing the window of length $t$ into $t/c$ segments, the probability of a output sequence only depends on the counts of outputs from the larger alphabet and follows a multinomial distribution. The total number of possible counts is at most $\left(\frac{2t}{c}\right)^{m^c}$ as each of the $m^c$ outputs can have a count from 0 to $t/c$. Therefore windows of length $t$ give at most $\left(\frac{2t}{c}\right)^{m^c}$ independent measurements, hence the window length has to be at least $O(n^{1/m^c})$ for the model to be identifiable as an HMM has $O(n^2 + nm)$ independent parameters. $\quad\square$

## C  Additional proofs for Section 3: Learnability results

**Assumptions for learning HMMs efficiently:**

For some fixed constants $c_1, c_2, c_3 > 1$, the HMM should satisfy the following properties for some $c > 0$:

1. *Transition matrix is well-conditioned:* Both $T$ and the transition matrix $T'$ of the time reversed Markov Chain are well-conditioned in the $\ell_1$-norm: $\sigma_{\min}^{(1)}(T), \sigma_{\min}^{(1)}(T') \geq 1/m^{c/c_1}$

2. *Transition matrix does not have short cycles:* For both $T$ and $T'$, every state visits at least $10 \log_m n$ states in $15 \log_m n$ time except with probability $\delta_1 \leq 1/n^c$.

3. *All hidden states have small "degree":* There exists $\delta_2$ such that for every hidden state $i$, the transition distributions $T_i$ and $T_i'$ have cumulative mass at most $\delta_2$ on all but $d$ states, with $d \leq m^{1/c_2}$ and $\delta_2 \leq 1/n^c$. Hence this is a soft "degree" requirement.

4. *Output distributions are random and have small support:* There exists $\delta_3$ such that for every hidden state $i$ the output distribution $O_i$ has cumulative mass at most $\delta_3$ on all but $k$ outputs, with $k \leq m^{1/c_3}$ and $\delta_3 \leq 1/n^c$. Also, the output distribution $O_i$ is randomly chosen from the simplex on these $k$ outputs.

**Theorem 2.** *If an HMM satisfies the above conditions, then with high probability over the choice of $O$, the parameters of the HMM are learnable to within additive error $\epsilon$ with observations over windows of length $2\tau + 1, \tau = 15 \log_m n$, with the sample complexity $poly(n, 1/\epsilon)$.*

*Proof sketch.* We refer the reader to Section 2.3 for the high level idea. Here, we provide a proof sketch for a much simpler case than that considered in Theorem 2. Recall that our main goal is to show that the likelihood matrix $A$ is well-conditioned. Assume for simplicity that the output distribution of each hidden state is deterministic so the output distribution only has support on one of

the $m$ character. The character on which the output distribution of each hidden state is supported is assigned independently and uniformly at random from the output alphabet. Also assume that $\delta_1, \delta_2, \delta_3$ in the conditions for Theorem 2 are zero. Our proof steps are roughly as follows–

1. Consider two random walks $m_1$ and $m_2$ on $T$ starting at disjoint sets of hidden states at time 0.

2. We first show that any two sample paths of a random walk on $T$ over $\tau = 15 \log_m n$ time steps, both of which visit $10 \log_m n$ different states in $\tau$ time steps but never meet in $\tau$ time steps, emit a different sequence of observations with high probability over the randomness in $O$.

3. Using the fact that the degree of each hidden state is small, we perform a union bound over all possible sample paths to show that with high probability over the choice of $O$, any two sample paths which do not meet in $\tau$ time steps emit a different sequence of observations.

4. Consider any 2 sample paths $s_1$ and $s_2$ corresponding to the random walks $m_1$ and $m_2$ which emit the same sequence of observations $w$ over $\tau$ time steps. By point 3 above, they must meet at some time $t$. If the probability of emitting $w$ under the random walks $m_1$ and $m_2$ are $p_1$ and $p_2$ respectively and $p_1 > p_2$, then we show that $(p_1 - p_2)$ of the probability mass in $m_1$ can be coupled with $m_2$ as these sample paths intersect sample paths from $m_2$. This is the core of the argument. Also refer to Fig. 5.

5. Hence if the probability of emitting a sequence of observations $w$ under the random walks $m_1$ and $m_2$ is very similar for every sequence $w$, then there is a very good coupling of the random walks, which implies that the total variational distance between the distribution of the random walks after $\tau$ time steps must be small. But this is a contradiction as $(\sigma_{\min}^{(1)}(T))^\tau$ is large. The contradiction stems from the fact that the $\ell_1$ distance between $m_1$ and $m_2$ at time 0 is one (as they start at disjoint starting states) and hence the distance at time $\tau$ is at least $(\sigma_{\min}^{(1)}(T))^\tau$.

$\square$

Figure 5: Consider two random walks $m_1$ and $m_2$ for 4 time steps with disjoint starting states and with sample paths $s_1$ and $s_2$ which visits the states *{a,b,c,d}* and *{e,b,c,f}* at times $\{0, 1, 2, 3\}$ respectively. We show that any two sample paths that have the same output distribution must be at the same hidden state at some time step. For example, here $s_1$ and $s_2$ are simultaneously at states *b* and *c*. This means that the probability mass in the two random walks can be coupled, hence the variational distance between the random walks $m_1$ and $m_2$ must be small at the end. But, this cannot be the case as $T$ is well-conditioned. Hence most sample paths of $m_1$ and $m_2$ must have different output distributions, which means that random walks $m_1$ and $m_2$ which start at disjoint states must have different output distributions, which implies that $A$ is well-conditioned.

*Proof.* Let the window length $N = 2t + 1$ where $t = 15 \log_m n$. We will prove the theorem for $c_1 = 20, c_2 = 16$ and $c_3 = 10$, these can be modified for different tradeoffs. By Lemma 1 from Bhaskara et al. [6], the simultaneous diagonalization procedure in Algorithm 1 needs $\text{poly}(n, 1/\beta, \kappa, 1/\epsilon)$ samples to ensure that the decomposition is accurate to an additive error $\epsilon$, with $\beta$ and $\kappa$ defined below.

**Lemma 1.** *[6] Suppose we are given a tensor $M + E \in \mathbb{R}^{m \times n \times p}$ with the entries of $E$ being bounded by $\epsilon = poly(1/\kappa, 1/n, 1/\beta)$ and $M$ has a decomposition $M = \sum_{i=1}^{R} A_i \otimes B_i \otimes C_i$ which satisfies-*

1. *The condition numbers $\kappa(A), \kappa(B) \leq \kappa$.*

2. *The column vectors of $C$ are not close to parallel: for all $i \neq j$, $\left\| \frac{w_i}{\|w_i\|_2} - \frac{w_j}{\|w_j\|_2} \right\|_2 \geq \beta$*

3. *The decompositions are bounded: for all $i$, $\|u_i\|_2, \|v_i\|_2, \|w_i\|_2 \leq K$*

*then the simultaneous decomposition algorithm recovers each rank one term in the decomposition of $M$ (up to renaming), within an additive error of $\epsilon$.*

Lemma 2 (proved in Section C.1) shows that the observation matrix $O$ satisfies condition 2 in Lemma 1 with $\beta = 1/n^{6.5}$.

**Lemma 2.** *If each column of the observation matrix is uniformly random on a support of size $k$, then $\|\frac{O_i}{\|O_i\|_2} - \frac{O_j}{\|O_j\|_2}\|_2 \geq 1/n^{6.5}$ with high probability over the choice of $O$.*

Note that as $A, B$ and $O$ are all stochastic matrices, all the factors have $\ell_2$ norm at most 1, therefore condition 3 in Lemma 1 is satisfied with $K = 1$. Hence if we show that the condition numbers $\kappa(A), \kappa(B) \leq poly(n)$, then each rank one term $A_i \otimes B_i \otimes C_i$ can be recovered up to an additive error of $\epsilon$ with $poly(n, 1/\epsilon)$ samples. As $\pi_i$ is at least $1/poly(n)$ for all $i$, this implies that $A, B, C$ can be recovered with additive error $\epsilon$ with $poly(n, 1/\epsilon)$ samples. As $O$ can be recovered from $C$ by normalizing $C$ to have unit $\ell_1$ norm, hence $O$ can be recovered up to an additive error $\epsilon$ with $poly(n, 1/\epsilon)$ samples.

If the estimate $\hat{A}$ of $A$ is accurate up to an additive error $\epsilon$, then the estimate $\hat{A}^{(t-1)}$ of $A^{(t-1)}$ is accurate up to an additive error $O(m\epsilon)$. Therefore, if the estimate $\hat{O}$ of $O$ is also accurate up to an additive error $\epsilon$ then the estimate of $\hat{A}' = (\hat{O} \odot \hat{A}^{(t-1)})$ of $A'$ is accurate up to an additive error $O(n\epsilon)$. Further, if the matrix $A'$ also has condition number at most $poly(n)$, then the transition matrix $T$ can be recovered using Algorithm 1 with up to an additive error of $\epsilon$ with $poly(n, 1/\epsilon)$ samples. Hence we will now show that the condition number $\kappa(A) \leq poly(n)$. The proof for an upper bound for $\kappa(A')$ and $\kappa(B)$ follows by the same argument.

Define the $(1 - \delta)$-support of any distribution $p$ as the set such that $p$ has mass at most $\delta$ outside that set. For convenience, define $\delta = \max\{\delta_1, \delta_2, \delta_3\}$ in the conditions for Theorem 2. We will find the probability that the Markov chains only undertakes transitions belonging to the $(1 - \delta)$-support of the current hidden state at each time step. As the probability of transitioning to any state outside the $(1 - \delta)$-support of the current hidden state is at most $\delta$, and the transitions at each time step are independent conditioned on the hidden state, the probability that the Markov chains only undertakes transitions belonging to the $(1 - \delta)$-support of all hidden state for each of the $n$ time steps is at least $(1 - \delta)^t \geq 1 - 2t\delta$. [2]

By the same argument, the probability of a sequence of hidden states always emitting an output which belongs to the $(1 - \delta)$-support of the output distribution of the hidden state at that time step is at least $(1 - \delta)^t \geq 1 - 2t\delta$.

We now show that two sequences of hidden states which do not intersect have large distance between their output distributions. The output alphabet over a window of size $\tau$ has size $K = m^t$. Let $\{a_i, i \in [K]\}$ be the set of all possible output in $t$ time steps. For any sequence $s$ of hidden states over a time interval $t$, define $o_s$ to be the vector of probabilities of output strings conditioned on any sequence of hidden states $s$, hence $o_s \in \mathbb{R}^K$ and the first entry of $o_s$ equals $P(a_1|s)$. Lemma 3 (proved in Section C.2) shows that the output distributions of sample paths which do not meet in $\tau$ time steps is large with high probability.

**Lemma 3.** *Let $s_i$ and $s_j$ be two sequences of $t$ hidden states which do not intersect. Also assume that $s_i$ and $s_j$ have the property that the output distribution at every time step corresponds to the $(1 - \delta)$ support of the hidden state at that time step. Let $o_{s_i}$ be the vector of probabilities of output*

*strings conditioned on any sequence of hidden states $s_i$. Also, assume that $s_i$ and $s_j$ both visit at least $(1 - \alpha)n$ different hidden states. Then,*

$$P\Big[\|o_{s_i} - o_{s_j}\|_1 = 1\Big] \geq \Big(\frac{4m^2}{d}\Big)^{0.5(1-\alpha)t}$$

Note that $\alpha \leq 1/3$ according to our condition 3 of Theorem 2. Also, as $k < m^{1/10}$, therefore $\Big(\frac{4k^2}{m}\Big)^{t/3} \leq (4/m^{4/5})^{t/3}$. Now consider the set $\mathcal{M}$ of all sequences with the property that the transition at every time step corresponds to the $(1 - \delta)$-support of the hidden state at that time step. As the $(1 - \delta)$-support of every hidden state has size at most $d$, $|\mathcal{M}| \leq nd^t$. Now for a sequence $s_i$, consider the set $\mathcal{M}_{s_i}$ of all sequences $s_j$ which do not intersect that sequence. By a union bound-

$$P\Big[\|o_{s_i} - o_{s_j}\|_1 = 1 \; \forall s_j \in \mathcal{M}_{s_i}\Big] \geq 1 - nd^t (4/m^{4/5})^{t/3}$$

Now doing a union bound over all sequences $s_i$,

$$
\begin{aligned}
P\Big[\|o_{s_i} - o_{s_j}\|_1 = 1 \; \forall s_i; s_j \in \mathcal{M}_{s_i}\Big] &\geq 1 - n^2 d^{2t} (4/m^{4/5})^{t/3} \\
&\geq 1 - \frac{n^2 m^{30 \log_m n/16} 4^{5 \log_m n}}{m^{4 \log_m n}} \\
&\geq 1 - \frac{n^{5/\log_4 m} n^{31/8}}{n^4}
\end{aligned}
$$

which is $1 - o(1)$. Hence every non-intersecting sequence of states in $\mathcal{M}$ has a different emission with high probability over the random assignment.

Using this property of $O$, we will bound the condition number of $A$. We will lower bound $\sigma_{\min}^{(1)}(A)$. As $\sigma_{\min}^{(2)}(A) \geq \sigma_{\min}^{(1)}(A)/\sqrt{n}$ and $\sigma_{\max}^{(2)}(A) \leq \sqrt{n}$, $\kappa(A) \leq n\sigma_{\min}^{(1)}(A)$.

Consider any $x \in \mathbb{R}^n$ such that $\|x\|_1 = 2$. We aim to show that $\|Ax\|_1$ is large. Let $x^+$ be the vector of all positive entries of $x$ i.e. $x_i^+ = \max(x_i, 0)$ where $x_i$ denotes the $i$th entry of vector $x$. Similarly, $x^-$ is the vector of all negative entries of $x$, $x_i^- = \max(-x_i, 0)$. We will find a lower bound for $\|Ax^+ - Ax^-\|_1$. Note that because $A$ is a stochastic matrix, there exists and $x$ that minimizes $\|Ax\|_1$ and has $1^T x = 0$. Hence we will assume without loss of generality that $1^T x = 0$, and hence $x^+$ and $x^-$ are valid probability distributions. Hence our goal is to show that for any two initial distributions $x^+$ and $x^-$ which have disjoint support, $\|Ax^+ - Ax^-\|_1 \geq 1/\text{poly}(n)$ which implies $\|Ax^+ - Ax^-\|_1 \geq 1/\text{poly}(n)$ for any $x \in \mathbb{R}^n$ such that $\|x\|_1 = 2$.

Note that $\|T^n x\|_1 > (\sigma_{\min}^{(1)}(T))^t$. Hence the distributions $x^+$ and $x^-$ do not couple in $n$ time steps with probability at least $(\sigma_{\min}^{(1)}(T))^t$. Note that $Ax^+$ is a vector where the $i$th entry is the probability of observing string $a_i$ with the initial distribution $x^+$. Let $\mathcal{U}$ denote the set of all $n^t$ sequences of hidden states over $t$ time steps. We exclude all sequences which do not visit $(1 - \alpha)t$ hidden states in $t$ time steps and sequences where at least one transition is outside the $(1 - \delta)$ support of the current hidden state. Let $\mathcal{S}$ be the set of all sequences of hidden states which visit at least $(1 - \alpha)n$ hidden states and where each transition is in the $(1 - \delta)$ support of the current hidden state. Let $\mathcal{X} = \mathcal{U} \backslash \mathcal{S}$. Note that for any distribution $p$, $Ap$ can be written as $Ap = \sum_{s \in \mathcal{X}} P(s|p)o_s$ where $P(s|p)$ is the probability of sequence $s$ with initial distribution $p$. Recall that $o_s \in \mathbb{R}^K$ is the vector of probabilities of outputs over the $K = m^t$ size alphabet conditioned on the sequence of hidden states $s$. Taking $p$ to be $e_i$, each column $c_i$ of $A$ can be expressed as $c_i = \sum_{s \in \mathcal{X}} P(s|e_i)o_s$. Restricting to sequences in $\mathcal{S}$, we define two new matrices $A_1$ and $A_2$, with the $i$th column $c_{i,1}$ of $A_1$ defined as $c_{i,1} = \sum_{s \in \mathcal{S}} P(s|e_i)o_s$ and the $i$th column $c_{i,2}$ of $A_2$ is analogously defined as $c_{i,2} = \sum_{s \in \mathcal{X}} P(s|e_i)o_s$. Note that $A = A_1 + A_2$. Also, as every hidden state visits less than $(1 - \alpha)t$ hidden states over $t$ time steps with probability at most $\delta$, and the probability of taking at least one transition outside the $(1 - \delta)$ support is at most $t\delta$, therefore $\sum_{s \in \mathcal{X}} P(s|e_i) \leq \delta + t\delta$ for all $i$. Hence $\|A_2 p\|_1 \leq \delta + t\delta$ for any vector $p$ with $\|p\|_1 = 1$.

We will now lower bound $\|A_1 x\|_1$. The high level proof idea is that if the two initial distributions do not couple then some sequences of states have more mass in one of the two distributions. Then, we use the fact that most sequences of states comprise of many distinct states to argue that two sequences of hidden states which do not intersect lead to very different output distributions. We combine these two observations to get the final result.

Consider any $O$ which satisfies the property every non-intersecting sequence of states has a different emission. We divide the output distribution $o_s$ of any sequence $s$ into two parts, $o_s^{(1)}$ and $o_s^{(2)}$. Define the $(1-\delta)$-support of $o_s$ as the set of possible outputs obtained when the emission at each time step belongs to the $(1-\delta)$ support of the output at that time step. We define $o_s^{(1)}$ as $o_s$ restricted to the $(1-\delta)$ support of $o_s$ and $o_s^{(2)}$ to be the residual vectors such that $o_s = o_s^{(1)} + o_s^{(2)}$. Note that $\|o_s^{(2)}\|_1$ is at most $t\delta$ for any sequence $s_i$.

Using our decomposition of the output distributions $o_s$ we will decompose $A_1$ as the sum of two matrices $A_1'$ and $A_1''$. The $i$th column $c_i'$ of $A_1'$ is defined as $c_i' = \sum_{s \in \mathcal{S}} P(e_i|s)o_s^{(1)}$. Similarly, the $i$th column $c_i''$ of $A_1''$ is defined as $c_i'' = \sum_{s \in \mathcal{S}} P(e_i|s)o_s^{(2)}$. As $\|o_{s_i}^{(2)}\|_1 \le t\delta$ for every sequence $s_i$, $\|A_1''p\|_1 \le t\delta$ for any vector $p$ with $\|p\|_1 = 1$.

We will further divide every sequence $s$ into a set of augmented sequences $\{s_{a_1}, s_{a_2}, \cdots, s_{a_K}\}$. Recall that $K = d^t$ is the size of the output space in $t$ time steps and $\{a_i, i \in [K]\}$ is the set of all possible output in $t$ time steps. The probability of sequence $s_{a_i}$ is the product of the probability of sequence $s$ times the probability of the sequence $s$ emitting the observation $a_i$. Hence, the probability of each augmented sequence equals the product of the probability of making the corresponding transition at each time step and emitting the corresponding observation at that time step.

For each output string $a_i$, there is a set of augmented sequences which have non-zero probability of emitting $a_i$. Consider the first string $a_1$. Let $\mathcal{S}_1^+$ be the set of all augmented sequences from the $x^+$ distribution with non-zero probability of emitting $a_1$, similarly let $\mathcal{S}_1^-$ be the set of all augmented sequences from the $x^-$ distribution with non-zero probability of emitting $a_1$. For any set of augmented sequences $\mathcal{S}$, let $|\mathcal{S}|$ denote the total probability mass of augmented sequences in $\mathcal{S}$.

We now show that any assignment of augmented sequences to outputs $a_i$ induces a coupling for two Markov chains $m^+$ and $m^-$ starting with initial distributions $x^+$ and $x^-$ respectively and having transition matrix $T$. We denote two sequences $s^+$ and $s^-$ as having coupled if they meet at some time step $u$ and traverse the Markov chain together after $u$, and the probability of $s^+$ equals the probability of $s^-$. Let $\mathcal{C}$ denote some coupling, and $\bar{\mathcal{C}}$ denote the total probability mass on sequences which have not been coupled under $\mathcal{C}$. Note that the total variational distance between the distribution of hidden states at time step $n$ from the starting distribution $x^+$ and $x^-$ satisfies

$$\|T^t x^+ - T^t x^-\|_1 \le \left|\bar{\mathcal{C}}\right|$$

for any coupling $\mathcal{C}$ with uncoupled mass $\bar{\mathcal{C}}$. This follows because all coupled sequences have the same distribution at time $t$, hence the distance between the distributions is at most the mass on the uncoupled sequences.

We claim that the sets of augmented sequences $\mathcal{S}_1^+$ and $\mathcal{S}_1^-$ can be coupled with residual mass $|(A_1'x^+)_1 - (A_1'x^-)_1|$, where $(A_1'x^+)_1$ denotes the first entry of $A_1'x^+$. To verify this, first assume without loss of generality that $(A_1'x^+)_1 > (A_1'x^-)_1$. Let $P(a_1, h_i|x^+)$ be the probability of outputting $a_1$ in $t$ time steps and being at hidden state $h_i$ at time 0, given the initial distribution $x^+$ at time 0. This is the sum of the probability of augmented sequences in $\mathcal{S}_1^+$ which start from hidden state $h_i$. Any coupling of the augmented sequences also induces a coupling of the probability masses $p^+ = \{P(a_1, h_i|x^+), i \in [n]\}$ and $p^- = \{P(a_1, h_i|x^-), i \in [n]\}$. We will show that all of the probability mass $p^-$ can be coupled. Consider a simple greedy coupling scheme $\mathcal{C}_1$ which picks a starting state $i$ in $p^-$, traverses along the transitions from the state $i$ and couples as much probability mass on sequences starting from $i$ as possible whenever it meets a sequence from $p^+$, and repeats for all starting states, till there are no more sequences which can be coupled. We claim that the algorithm terminates when all of the probability mass in $p^-$ has been coupled. We prove by contradiction. Assume that there exists some probability mass $p^-$ which has not been coupled in the end. There must be also be some probability mass $p^+$ which has not been coupled. But all augmented sequences starting from hidden state $i$ meet with all sequences from hidden state $j$, so this means that more probability mass from $p^-$ can be coupled to $p^+$. This contradicts the assumption that there are no more augmented sequences which can be coupled. Hence all of the probability mass in $p^-$ has been coupled when the greedy algorithm terminates. Hence coupling $\mathcal{C}_1$ has residual mass $\left|\bar{\mathcal{C}_1}\right|$ at most $|(A_1'x^+)_1 - (A_1'x^-)_1|$.

Now, consider all outputs $a_i$ and couplings $\mathcal{C}_i$. Let $\mathcal{C} = \cup_i \mathcal{C}_i$. As our argument only couples the augmented sequences the mass $\|A_1''x^+\|_1 + \|A_1''x^-\|_1 + \|A_2x^+\|_1 + \|A_2x^-\|_1$ is never uncoupled. The total uncoupled mass is,

$$\left|\bar{\mathcal{C}}\right| = \sum_i \left|\bar{\mathcal{C}}\right| + \|A_1''x^+\|_1 + \|A_1''x^-\|_1 + \|A_2x^+\|_1 + \|A_2x^-\|_1$$

$$\implies \left|\bar{\mathcal{C}}\right| \leq \|A_1'x^+ - A_1'x^-\|_1 + 6t\delta$$

$$\implies \|T^t x^+ - T^t x^-\|_1 \leq \|A_1'x^+ - A_1'x^-\|_1 + 6t\delta$$

Note that $\|T^t x^+ - T^t x^-\|_1 \geq (\sigma_{\min}^{(1)}(T))^t$. By condition 2 in Theorem 2, $\sigma_{\min}^{(1)}(T) \geq 1/m^{c/20}$ therefore $\|T^t x^+ - T^t x^-\|_1 \geq 1/n^{3c/4}$. Note that $\|Ax\|_1 \geq \|A_1'x\|_1 - \|A_1''x\|_1 - \|A_2x\|_1 \geq 1/n^{3c/4} + 10t\delta$. As $\delta \leq 1/n^c$, therefore $\|Ax\|_1 \geq 1/n^{3c/4} - 0.5/n^{3c/4} \geq 0.5/n^{3c/4}$. $\qquad\square$

We also mention the following corollary of Theorem 2. Corollary 1 is defined in terms of the minimum singular value of the matrix, $\sigma_{\min}^{(2)}(T)$, and is a slightly weaker but more interpretable version of Theorem 2. The conditions are the same as in Theorem 2 with different bounds on $\delta_1, \delta_2, \delta_3$.

**Corollary 1.** *If an HMM satisfies $\delta_1, \delta_2, \delta_3 \leq 1/n^2$ and $\sigma_{\min}^{(2)}(T), \sigma_{\min}^{(2)}(T') \geq 1/m^{1/20}$ then with high probability over the choice of O, the parameters of the HMM learnable to within additive error $\epsilon$ with observations over windows of length $2\tau + 1, \tau = 15\log_m n$ with the sample complexity being $poly(n, 1/\epsilon)$.*

On a side note, though $\sigma_{\min}^{(2)}(T)$ is much more easier to interpret than $\sigma_{\min}^{(1)}(T)$ – for example, if the transition matrix is symmetric then the singular values are the same as the eigenvalues, and the eigenvalues of a matrix have well-known connections to the properties of the underlying graph, but, because $T$ is a stochastic matrix and all columns have unit $\ell_1$ norm, the $\ell_1$ norm seems better suited to measuring the gain of the matrix. For example, if the transition matrix has a single state which transitions to $d$ states with equal probability, then $\sigma_{\min}^{(2)}(T) \leq 1/\sqrt{d}$ by choosing the vector which has unit mass on that state, hence if $d$ is large then $\sigma_{\min}^{(2)}(T)$ is always small even when the transition matrix is otherwise well-behaved.

### C.1 Proof of Lemma 2

**Lemma 2.** *If each column of the observation matrix is uniformly random on a support of size $k$, then $\|\frac{O_i}{\|O_i\|_2} - \frac{O_j}{\|O_j\|_2}\|_2 \geq 1/n^{6.5}$ with high probability over the choice of O.*

*Proof.* The proof is in two parts. In the first part we show that $\|O_i - O_j\|_1 \geq 1/n^5$ with high probability. In the second part we show that $\left\|\frac{O_i}{\|O_i\|_2} - \frac{O_j}{\|O_j\|_2}\right\|_2 \geq 1/n^{6.5}$ if $\|O_i - O_j\|_1 \geq 1/n^5$.

We will show that $\|O_i - O_j\|_1 \geq 1/n^5$ with high probability in the smoothed sense, which implies that it is also true in our model where they are chosen uniformly on a small support. Consider any two distributions $O_i$ and $O_j$. Let $O_i$ have largest mass on some state $f_i$ and next largest mass on another state $g_i$. Similarly, let $O_j$ have largest mass on some state $f_j$ and next largest mass on another state $g_j$. Let $x_1$ and $x_2$ be random variables uniformly distributed in $[0, 1/n^2]$. Say we perturb $O_i$ by subtracting $x_1$ from the probability of $f_i$ and adding $x_1$ to the probability of $g_i$. Similarly, say we perturb $O_j$ by subtracting $f_2$ from the probability of $u_j$ and adding $x_2$ to the probability of $g_j$. With probability $1/n^3$ over the choice of $x_1$ and $x_2$, $|x_1 - x_2| \geq 1/n^5$, which implies $\|O_i - O_j\|_1 \geq 1/n^5$. Therefore by a union bound over all pairs $O_i$ and $O_j$, $\|O_i - O_j\|_1 \geq 1/n^5$ with high probability.

We now show that $\left\|\frac{O_i}{\|O_i\|_2} - \frac{O_j}{\|O_j\|_2}\right\|_2 \geq 1/n^{6.5}$ when $\|O_i - O_j\|_1 \geq 1/n^5$. We prove the contrapositive via the following Lemma.

**Lemma 4.** *For any two vectors $v_1$ and $v_2$, $\left\|\frac{v_1}{\|v_1\|_1} - \frac{v_2}{\|v_2\|_1}\right\|_1 < 1/n^5$ if $\left\|\frac{v_1}{\|v_1\|_2} - \frac{v_2}{\|v_2\|_2}\right\|_2 \leq 1/n^{6.5}$.*

*Proof.* As the claim is scale invariant, assume $\|v_1\|_2 = 1$ and $\|v_2\|_2 = 1$. As $\|v_1 - v_2\|_2 \leq 1/n^{6.5}$, therefore $\|v_1 - v_2\|_1 \leq 1/n^6$. Therefore $|\|v_1\|_1 - \|v_2\|_1| \leq 1/n^6$. Let $\|v_1\|_1 = x$, where $x \geq 1$,

$\|v_2\|_1 = x + \delta$, where $|\delta| \le 1/n^6$.

$$\left\|\frac{v_1}{\|v_1\|_1} - \frac{v_2}{\|v_2\|_1}\right\|_2 = \left\|\frac{v_1}{x} - \frac{v_2}{x + \delta}\right\|_2$$

$$= \frac{1}{x}\left\|v_1 - \frac{v_2}{1 + \delta/x}\right\|_2$$

$$= \frac{1}{x}\left\|v_1 - v_2(1 + \epsilon/x)\right\|_2$$

for some $\epsilon$ with $|\epsilon| \le 2|\delta|$. Therefore using the triangle inequality,

$$\left\|\frac{v_1}{\|v_1\|_1} - \frac{v_2}{\|v_2\|_1}\right\|_2 \le \frac{1}{x}\|v_1 - v_2\|_2 + \frac{\epsilon}{x}\|v_2\|_2$$

$$\le 1/n^6 + 2/n^6 < 1/n^{5.5}$$

$$\implies \left\|\frac{v_1}{\|v_1\|_1} - \frac{v_2}{\|v_2\|_1}\right\|_1 < 1/n^5$$

$\square$

Using Lemma 4, it follows that $\left\|\frac{O_i}{\|O_i\|_2} - \frac{O_j}{\|O_j\|_2}\right\|_2 > 1/n^{6.5}$ when $\|O_i - O_j\|_1 \ge 1/n^5$

$\square$

## C.2  Proof of Lemma 3

**Lemma 3.** *Let $s_i$ and $s_j$ be two sequences of $t$ hidden states which do not intersect. Also assume that $s_i$ and $s_j$ have the property that the output distribution at every time step corresponds to the $(1 - \delta)$ support of the hidden state at that time step. Let $o_{s_i}$ be the vector of probabilities of output strings conditioned on any sequence of hidden states $s_i$. Also, assume that $s_i$ and $s_j$ both visit at least $(1 - \alpha)n$ different hidden states. Then,*

$$P\left[\|o_{s_i} - o_{s_j}\|_1 = 1\right] \ge \left(\frac{4m^2}{d}\right)^{0.5(1-\alpha)t}$$

*Proof.* For the output distributions corresponding to sequence $s_i$ and $s_j$ to not have disjoint supports, the hidden state visited by the two sequences at every time step must have overlapping output distributions. The probability of any pair of hidden states having overlapping support is at most $1 - (1 - 2k/m)^k \le 4k^2/m$.

Consider the graph $G$ on $n$ nodes, where we connect node $u$ and node $v$ if sequence $s_i$ and $s_j$ are simultaneously at hidden states $v$ and $v$ at some time step. As each sequence visits at least $(1 - \alpha)t$ different hidden states, at least $(1 - \alpha)t$ nodes in $G$ have non-zero degree. Consider any connected component $C$ of the graph with $p$ nodes. Note that the probability of the output distributions corresponding to each edge in the connected component $C$ to be overlapping equals the probability of the output distributions of each node in $C$ to be overlapping. This is at most $\left(\frac{4k^2}{m}\right)^{p-1}$. Now, let there be $M$ connected components in graph each of which has $p_i$ nodes. The probability of the sequences having the same support is at most

$$P\left[\|o_{s_i} - o_{s_j}\|_1 < 1\right] \le \Pi_{i=1}^M \left(\frac{4k^2}{m}\right)^{p_i - 1}$$

$$\le \left(\frac{4k^2}{m}\right)^{\sum_i (p_i - 1)} \le \left(\frac{4k^2}{m}\right)^{(1-\alpha)t - M}$$

$$\le \left(\frac{4k^2}{m}\right)^{(1-\alpha)t/2}$$

where the last step follows because $\sum_i p_i$ is the number of nodes which have non-zero degree, which is at least $\ge (1 - \alpha)t$, and as every connected component has at least 2 nodes, there are at most $(1 - \alpha)t/2$ connected components, therefore $M \le (1 - \alpha)t/2$. $\square$

# D   Proof of lower bound for dense HMMs

**Theorem 1.** *Consider the class of HMMs with $n$ hidden states and $m$ outputs and $m = polylog(n)$ with the transition matrix chosen to be a $d$-regular graph, with $d = n^\epsilon$ for some $\epsilon > 0$. Then at least $\Omega(nd)$ bits of information are needed to specify the choice of the transition matrix. However, if the observation matrix $O$ is randomly chosen such that the columns of $O$ are chosen independently and $\mathbb{E}[O_{ij}] = 1/m$ for all $i, j$, then the number of bits of information contained in polynomially many samples over a window of length $N = poly(n)$ is at most $\tilde{\mathcal{O}}(n)$, with high probability over the choice of $O$, where the $\tilde{\mathcal{O}}$ notation hides polylogarithmic factors in $n$.*

*Proof sketch.* The proof consists of two steps–in the first step we show that the information contained in polynomially many observations over windows of length $\tau = \lfloor \log_m n \rfloor$ is not sufficient to learn the HMM. The proof of this part relies on a counting argument and a lower bound on the number of random regular graphs with a given degree. We then show that the information contained in polynomial samples over longer windows is not much larger than the information contained in polynomial samples over a window length of $\tau$. This is the main technical part, and we need to show that the hidden state at time $0$ does not have much influence on the hidden state at time $t$, conditioned on the outputs from time $0$ to $t$. The conditioning makes this tricky, as the probabilities of the hidden states no longer evolve under the transition matrix of the Markov chain. We get around this by showing that the probability of the hidden states after conditioning on the observations evolves under a time-inhomogeneous Markov chain, and the transition matrices at every time step are related to the outputs from time $1$ to $t$ and the original transition matrix. We analyze the spectrum of the time-inhomogeneous transition matrices to show that the influence of the hidden state at time $0$ decays at every step and is small at time $t$.

We would like to point out that our techniques to prove the information theoretic lower bound appear to be generally useful for analyzing the influence of the hidden state at time $0$ on the hidden state at time $t$, conditioned on the outputs from time $0$ to $t$, This is a measure of how much value there is to observations before time $0$ for predicting the observation at time $t + 1$, conditioned on the intermediate observations from time $0$ to $t$. This is a natural notion of the memory of the output process. $\qquad\qquad\square$

*Proof.* By Shamir and Upfal [22] (also see Krivelevich et al. [17]), the number of $d$-regular graphs on $n$ vertices is at least $\binom{\binom{n}{2}}{nd/2} \exp(-nd^{0.5+\delta})$ for any fixed $\delta > 0$. This can be bounded from below as follows—

$$\binom{\binom{n}{2}}{nd/2} \exp(-nd^{0.5+\delta}) = \left(\frac{n-1}{d}\right)^{nd/2} \exp(-nd^{0.5+\delta})$$

$$\geq 2^{nd/2 - nd^{0.5+\delta}}$$

Hence the number of bits needed to specify a randomly chosen $d$-regular graph on $n$ vertices is at least $\Omega(nd)$.

Note that if we only get observations over a window of length $\tau = \lfloor \log_m n \rfloor$, and we obtain $poly(n)$ samples, then the total information in those samples is at most $\tilde{O}(n)$, where the $\tilde{O}$ hides polylogarithmic factors in $n$. This is because there are at most $m^\tau \leq n$ possible outputs, each of which can take $poly(n)$ different values as there are $poly(n)$ samples. We will now show that getting polynomially many samples over windows of length $N = poly(n)$ is equivalent to getting polynomially many samples over windows of length $\tau$.

For notational convenience, we will refer to $P[o_t = i]$, the probability of the output at time $t$ being $i$, as $P[o_t]$ whenever the assignment to the random variable is clear from the context. The probability of any sequence of outputs $\{o_1, o_2, \cdots, o_N\}$ can be written down as follows using chain rule,

$$P[o_1, o_2, \cdots, o_N] = P[o_1]P[o_2|o_1]P[o_3|o_1, o_2] \cdots P[o_N|o_1, \cdots, o_{N-1}]$$

$$= \Pi_{t=1}^{N} P[o_t|o_1, \cdots, o_{t-1}]$$

In order to prove that the probabilities of sequences of length $N$ can be well-approximated using sequences of length $\tau$, for $t > \tau$, we will approximate the probability $P[o_t|o_1, \cdots, o_{t-1}]$ by

$P[o_t|o_{t-\tau+1}, \cdots, o_{t-1}]$. Hence our estimate of the probability of sequence $\{o_1, o_2, \cdots, o_N\}$ is

$$\hat{P}[\{o_1, o_2, \cdots, o_N\}] = \Pi_{t=1}^{\tau} P[o_t|o_{(t-\tau+1)\vee 1}, \cdots, o_{t-1}]$$

where $a \vee b$ denotes $\max(a, b)$. If

$$\|P[o_t|o_1, \cdots, o_{t-1}] - P[o_t|o_{(t-\tau+1)\vee 1}, \cdots, o_{t-1}]\|_1 \leq \epsilon \qquad (2)$$

for all $t \leq N$ and assignments to $\{o_1, \cdots, o_t\}$, then

$$\left| P[o_1, o_2, \cdots, o_N] - \hat{P}[o_1, o_2, \cdots, o_N] \right| \leq O(\epsilon N)$$

for all assignments to $\{o_1, \cdots, o_N\}$. Hence the probabilities of windows of length $N$ can be estimated from windows of length $\tau$ up to an additive error of $O(\epsilon N)$. Therefore, if we can show that $\epsilon \leq o(1/\text{poly}(n))$, then given the true probabilities of windows of length $\tau$, it is possible to estimate the true probabilities over windows of length $N = \text{poly}(n)$ up to an inverse super-polynomial factor of $n$. Given empirical probabilities of windows of length $\tau$ up to an accuracy of $\delta$, it is possible to estimate the true probabilities over windows of length $N$ up to an error of $(\epsilon + \delta)N$. As $\epsilon N$ is inverse super-polynomial in $N$, getting $S$ samples over windows of length $N$ is equivalent to getting $\text{poly}(N, S)$ samples over windows of length $\tau$. Therefore, the information contained in polynomially many samples over windows of length $N$ is entirely contained in the polynomially many samples over windows of length $\tau$ (the polynomials would be different, but this does not concern us as we have shown that the information in polynomially many samples over windows of length $\tau$ is always $\tilde{O}(n)$). Hence the information contained in polynomially many samples over windows of length $N$ can be at most $\tilde{O}(n)$.

We will now prove Eq. 2. First, note that Eq. 2 can be written in terms of the probabilities of the hidden states as follows,

$$P[o_t = j|o_1, \cdots, o_t] = \sum_{i=1}^{n} P[o_t = j|h_t = i]P[h_t = i|o_1, \cdots, o_t]$$

Therefore, it is sufficient to show the following, which says that the distribution of the hidden states conditioned on the two observation windows is similar–

$$\|P[h_t|o_1, \cdots, o_{t-1}] - P[h_t|o_{(t-\tau+1)\vee 1}, \cdots, o_{t-1}]\|_1 \leq \epsilon \qquad (3)$$

for all $t \leq N$ and assignments to $\{o_1, \cdots, o_{t-1}\}$. Note that we do not need to worry about the case when $t \geq \tau$ as the observation windows under consideration are the same for both terms. We will shift our windows and fix $t = \tau$ to make notation easy, as the process is stationary this can be done without loss of generality. Hence we will rewrite Eq. 3 as follows, ignoring the cases when the terms are the same because $(t - \tau + 1) \vee 1 = 1$.

$$\left| P[h_\tau|o_1, \cdots, o_{\tau-1}] - P[h_\tau|o_z, \cdots, o_{\tau-1}] \right| \leq \epsilon$$

for all $z \in [\tau - N + 1, 0]$.

Define the modified transition matrix $T^{(t)}$ as $T_{i,j}^{(t)} = P[h_{t+1} = j|h_t = i, o_{t+1}, \cdots, o_{\tau-1}]$. For any $s \in [0, \tau]$, we claim that,

$$P[h_s|o_z, \cdots, o_{\tau-1}] = \left( \Pi_{t=1}^{s} T^{(t)} \right) P[h_0|o_z, \cdots, o_{\tau-1}]$$

for all $z \in [\tau - N + 1, 0]$. Therefore $T^{(t)}$ serves the role of the transition matrix at time $t$ in our setup. The proof of this follows from a simple induction argument on $s$. The base case $s = 0$ is clearly correct. Let the statement be true up to some time $p$. Then we can write,

$$P[h_{p+1}|o_z, \cdots, o_{\tau-1}] = \sum_i P[h_p = i, h_{p+1}|o_z, \cdots, o_{\tau-1}]$$

$$= \sum_i P[h_p = i|o_z, \cdots, o_{\tau-1}]P[h_{p+1}|h_p = i, o_z, \cdots, o_\tau]$$

$$= \sum_i P[h_{p+1}|h_p = i, o_{p+1}, \cdots, o_{\tau-1}]\left( \Pi_{t=1}^{p} T^{(t)} \right) P[h_0|o_z, \cdots, o_{\tau-1}]$$

$$= \left( \Pi_{t=1}^{p+1} T^{(t)} \right) P[h_0|o_z, \cdots, o_{\tau-1}]$$

where we could simplify $P[h_{p+1}|h_p = i, o_z, \cdots, o_{\tau-1}] = P[h_{p+1}|h_p = i, o_{p+1}, \cdots, o_{\tau-1}]$ as conditioned on the hidden state at time $p$, the observations at and before time $p$ do not affect the distribution of future hidden states. Therefore, our task now reduces to analyzing the spectrum of the time-inhomogeneous transition matrices $T^{(t)}$. The following Lemma does this.

**Lemma 5.** *For any $x$ with $\|x\|_2 = 1$ and $1^T x = 0$, $\|T^{(t)}x\|_2 \le \alpha + \lambda$ where $\alpha \le \sqrt{\frac{100m^3 \log^3 n}{2d}}$ and $\lambda < 3/\sqrt{d}$. Therefore if $d = n^\epsilon$ for some $\epsilon > 0$ and $m = polylog(n)$, then $\|T^{(t)}x\|_2 \le n^{-\epsilon_1}$ for some $\epsilon_1 > 0$.*

Given Lemma 5, we will show $\epsilon = o(1/\text{poly}(n))$. Let $p_1 = P[h_0|o_1, \cdots, o_{\tau-1}]$ and $p_2 = P[h_0|o_z, \cdots, o_{\tau-1}]$ for any $z < 1$. We can write,

$$P[h_0|o_1, \cdots, o_{\tau-1}] - P[h_0|o_z, \cdots, o_{\tau-1}] = \left(\Pi_{t=1}^{\tau} T^{(t)}\right)(p_1 - p_2)$$

Let $p_1 - p_2 = x$. Note that $\|x\|_2 \le 1$ and $1^T x = 0$. Furthermore, as $T^{(t)}$ is stochastic for every $t$, therefore $1^T\left(\Pi_{t=1}^{s} T^{(t)}\right)x$ is also 0 for every $s$. Hence we can use Lemma 5 to say,

$$\|P[h_0|o_1, \cdots, o_{\tau-1}] - P[h_0|o_t, \cdots, o_{\tau-1}]\|_2 \le (2(\alpha + \lambda))^\tau$$
$$\implies \|P[h_0|o_1, \cdots, o_{\tau-1}] - P[h_0|o_t, \cdots, o_{\tau-1}]\|_1 \le \sqrt{n}(\alpha + \lambda)^\tau$$
$$\le n^{-\log^\delta n}$$

for a fixed $\delta > 0$. This is superpolynomial in $n$, proving Theorem 1. We will now prove Lemma 5.

**Lemma 5.** *For any $x$ with $\|x\|_2 = 1$ and $1^T x = 0$, $\|T^{(t)}x\|_2 \le \alpha + \lambda$ where $\alpha \le \sqrt{\frac{100m^3 \log^3 n}{2d}}$ and $\lambda < 3/\sqrt{d}$. Therefore if $d = n^\epsilon$ for some $\epsilon > 0$ and $m = polylog(n)$, then $\|T^{(t)}x\|_2 \le n^{-\epsilon_1}$ for some $\epsilon_1 > 0$.*

*Proof.* We show that $T^{(t)}$ has a simple decomposition, $T^{(t)} = O^{(t)} T E^{(t)}$, where $O^{(t)}$ is a diagonal matrix with $O_{i,i}^{(t)} = P(o_t, \cdots, o_\tau|h_t = i)$ and $E^{(t)}$ is another diagonal matrix with $E_{i,i}^{(t)} = P[o_{t+1}, \cdots, o_\tau|h_t = i]$. This is because,

$$P[h_{t+1} = j|h_t = i, o_{t+1}, \cdots, o_\tau] = \frac{P[h_{t+1} = j|h_t = i]P[o_{t+1}, \cdots, o_\tau|h_{t+1} = j]}{\sum_j P[h_{t+1} = j|h_t = i]P[o_{t+1}, \cdots, o_\tau|h_{t+1} = j]}$$
$$= \frac{P[h_{t+1} = j|h_t = i]P[o_{t+1}, \cdots, o_\tau|h_{t+1} = j]}{P[o_{t+1}, \cdots, o_\tau|h_t = i]}$$

As $T$ is the normalized adjacency of a $d-$regular graph, the eigenvector corresponding to the eigenvalue 1 is the all ones vector, and all subsequent eigenvectors are orthogonal to the all ones vector. Also, the second eigenvalue and all subsequent eigenvalues of $T$ are at most $3/\sqrt{d}$, due to Friedman [11].

To analyze $O^{(t)}$ and $E^{(t)}$, we need to first derive some properties of the randomly chosen observation matrix $O$. We claim that,

**Lemma 6.** *Denote $x_{ij}$ to be the random variable denoting the probability of hidden state $i$ emitting output $j$. If $\{x_{ij}, i \in [n]\}$ are independent and $\mathbb{E}[x_{ij}] = 1/m$ for all $i$ and $j$, then for all outputs $j \in [m]$ and hidden states $i \in [n]$, $\left|P[o_{t+1} = j|h_t = i] - 1/m\right| \le \sqrt{6 \log n/(dm)}.$*

*Proof.* The result is a simple application of Chernoff bound and a union bound. Without loss of generality, let the set of neighbors of hidden state $i$ be the hidden states $\{1, 2, \cdots, d\}$. $P[o_{t+1} = j|h_t = i]$ is given by,

$$P[o_{t+1} = j|h_t = i] = (1/d)\sum_{k=1}^{d} x_{kj}$$

Let $X_{ij} = \sum_{k=1}^{d} x_{kj}$. Note that the $x_{kj}$'s are all independent and bounded in the interval $[0, 1]$ with $\mathbb{E}[x_{kj}] = 1/m$. Therefore we can apply Chernoff bound to show that,

$$P\left[|X_{ij} - d/m| \geq \sqrt{\frac{6d \log n}{m}}\right] \leq 2 \exp(-2 \log n) \leq 2/n^2$$

Therefore, $\left|P[o_{t+1} = j | h_t = i] - 1/m\right| \leq \sqrt{6 \log n/(dm)}$ with failure probability at most $1/n^2$. Hence by performing a union bound over all hidden states and outputs, with high probability $|P[o_{t+1} = j | h_t = i] - 1/m| \leq \sqrt{6 \log n/(dm)}$ for all all outputs $j \in [m]$ and hidden states $i \in [n]$. $\qquad\square$

Using Lemma 6, it follows that

$$P[o_{t+1}, \cdots, o_\tau | h_t = i] \in \left[\frac{1}{m^{\tau-t}}\left(1 - \sqrt{\frac{3m \log^3 n}{2d}}\right), \frac{1}{m^{\tau-t}}\left(1 + \sqrt{\frac{24m \log^3 n}{d}}\right)\right]$$

This is because using Lemma 6, conditioned on any hidden state at any time $t$,

$$P[o_{t+1} = j | h_t = i] \in \left[\left(\frac{1}{m} - \sqrt{\frac{6 \log n}{dm}}\right), \left(\frac{1}{m} + \sqrt{\frac{6 \log n}{md}}\right)\right]$$

for all outputs $j$, hidden states $i$ and $t \in [0, \tau]$ hence the probability of emitting the sequence of outputs $\{o_t, \cdots, o_\tau\}$ starting from any hidden state is at most

$$\left(\frac{1}{m} + \sqrt{\frac{6 \log n}{md}}\right)^{\tau-t} \leq \frac{1}{m^{\tau-t}}\left(1 + \sqrt{\frac{24m\tau^2 \log n}{d}}\right) \leq \frac{1}{m^{\tau-t}}\left(1 + \sqrt{\frac{24m \log^3 n}{d}}\right)$$

and similarly for the lower bound. Therefore $O_{i,i}^{(t)} = P[o_{t+1}, \cdots, o_\tau | h_{t+1} = i]$ can be bounded as follows

$$P[o_{t+1}, \cdots, o_\tau | h_{t+1} = i] = P[o_{t+1} | h_{t+1} = i]P[o_{t+2}, \cdots, o_\tau | h_{t+1} = i] \leq P[o_{t+2}, \cdots, o_\tau | h_{t+1} = i]$$

$$\implies O_{i,i}^{(t)} \leq \frac{1}{m^{\tau-t-1}}\left(1 + \sqrt{\frac{24m \log^3 n}{d}}\right) \forall i, t$$

We will now bound the entries of $\tilde{E}^{(t)}$. Note that

$$1/E_{i,i}^{(t)} = P[o_{t+1}, \cdots, o_\tau | h_t = i]$$

$$\implies \frac{1}{m^{\tau-t}}\left(1 - \sqrt{\frac{3m \log^3 n}{2d}}\right) \leq 1/E_{i,i}^{(t)} \leq \frac{1}{m^{\tau-t}}\left(1 + \sqrt{\frac{24m \log^3 n}{d}}\right)$$

$$\implies m^{\tau-t}\left(1 - \sqrt{\frac{24m \log^3 n}{2d}}\right) \leq E_{i,i}^{(t)} \leq m^{\tau-t}\left(1 + \sqrt{\frac{12m \log^3 n}{d}}\right)$$

We can cancel the factor of $m^{\tau-t-1}$ appearing in the numerator of the upper bound for $E_{ii}^{(t)}$ and denominator of the upper bound for $P[o_{t+1}, \cdots, o_\tau | h_{t+1} = i]$ by multiplying $O^{(t)}$ by $m^{\tau-t}$ and dividing $E^{(t)}$ by $m^{\tau-t}$. Let the normalized matrices be $\tilde{O}^{(t)}$ and $\tilde{E}^{(t)}$. Therefore,

$$\tilde{O}_{i,i}^{(t)} \leq \left(1 + \sqrt{\frac{24m \log^3 n}{d}}\right)$$

$$m\left(1 - \sqrt{\frac{24m \log^3 n}{2d}}\right) \leq \tilde{E}_{i,i}^{(t)} \leq m\left(1 + \sqrt{\frac{12m \log^3 n}{d}}\right)$$

Consider any $x$ such that $1^T x = 0$ and $\|x\|_2 = 1$. Let $\tilde{E}^{(t)} x = \alpha v + x_2$, where $1^T x_2 = 0$, $\|x_2\|_2 \leq 1$ and $v$ is the all ones vector normalized to have unit $\ell_2$-norm. We claim that $\alpha = v^T x \leq \sqrt{\frac{100m^3 \log^3 n}{2d}}$.

As $|\tilde{E}_{i,i} - \tilde{E}_{j,j}| \leq \sqrt{\frac{100m^3 \log^3 n}{2d}}$ for all $i \neq j$ and $1^T x = 0$, therefore

$$|v^T \tilde{E}^{(t)} x| \leq \frac{\max_{i,j} |\tilde{E}_{i,i} - \tilde{E}_{j,j}|}{\sqrt{n}} \|x\|_1$$

$$\leq \sqrt{\frac{100m^3 \log^3 n}{2dn}} \|x\|_1 \leq \sqrt{\frac{100m^3 \log^3 n}{2d}} \|x\|_2 \leq \sqrt{\frac{100m^3 \log^3 n}{2d}}$$

Therefore, $T\tilde{E}^{(t)} x = \alpha v + T x_2$. Let the second eigenvalue of $T$ be upper bounded by $\lambda$. As $1^T x_2 = 0$ therefore $\|T x_2\|_2 \leq \lambda$. Hence $\|T\tilde{E}^{(t)} x\|_2 \leq \alpha + \lambda$ for any $x$ with $1^T x = 0$ and $\|x\|_2 = 1$. Note that the operator norm of the matrix $\tilde{O}^{(t)}$ is at most 2 because $\tilde{O}^{(t)}$ is a diagonal matrix with each entry bounded by $\left(1 + \sqrt{\frac{24m \log^3 n}{d}}\right) \leq 2$. Therefore $\|\tilde{O} T \tilde{E}^{(t)} x\|_2 \leq 2(\alpha + \lambda)$ for any $x$ with $1^T x = 0$ and $\|x\|_2 = 1$. $\qquad\square$

$\square$

[Supplementary Material 2]