[Reviews · NeurIPS 2017]

Reviewer 1



The paper studies the learnability of HMMs in the setting when the output label size m is smaller than the number of states n. This setting is particularly tricky since the usual full-rank assumption that comes up in tensor decomposition based methods is not valid. This paper gives both algorithms and information-theoretic lower-bounds on the learnability of overcomplete HMMs, which clearly increases our state of understanding. 1. They show that HMMs with m=polylog(n) even for transition-matrix being a random dense regular graphs, can not be learning using poly(n) samples even when the window size is poly(n). 2. They show that overcomplete HMMs can be learned efficiently with window size O(log_m n), under some additional four conditions (well-conditioned-ness of T, no-short-cycles, sparsity of graph given by T and random sparse output distributions) Positives: 1. The lower bound is interesting and surprisingly strong. This is unlike the parity-based lower bounds in Mossel-Roch which are more computational by nature, and more similar to moment-based identifiability lower bounds known for Gaussian mixtures and multiview models (see Rabani-Swamy-Schulman). 2. This paper tries to understand in depth the effect of the structure of the transition matrix on learnability in the overcomplete setting. 3. The techniques developed in this paper are new and significantly depart from previous works. For instance, the algorithmic result using tensor decompositions departs significantly from previous techniques in analyzing the Khatri-Rao (KR) product. Analyzing overcomplete tensor decompositions and these KR products are challenging, and the only non-generic analysis I am aware of (by Bhaskara et al. for smoothed analysis) can not be used here since the different tensor modes are dependent. At the heart of this argument is an interesting coupling argument that shows that for any two different starting states, the random walk distributions after O(log_m n) steps will be significantly different if the observation matrices are random. This crucially depends on the well-conditioned-ness of T and no-short-cycle condition which they argue is needed for identifiability. 4. I like the discussion of assumptions section 2.4; while the paper assumes four conditions, and they give qualitative reasons for why most of them are necessary. Negatives: 1. The proofs are fairly dense (especially Theorem 1 , Lemma 3) and the error analysis is hand-wavy at places . Though I have no reason to suspect the correctness, it would be good to improve the readability of these proofs. 2. It is not clear why some of the conditions are necessary. For instance, the random sparse support assumption seems stronger than necessary. Similarly, while the lower bound example in Proposition 1 explains why an example with just short cycles is not identifiable, this doesn't explain why there should be no short-cycles in a connected, well-conditioned T. Summary: Overall, the paper presents a nice collection of results that significantly improves our understanding on overcomplete HMMs. I think the techniques developed here may be of independent interest as well. I think the paper is easily above the bar. Comments: The identifiability results for HMMs using O(log_m n ) should also be attributed to Allman-Mathias-Rhodes'2010. They already show that the necessary conditions are true in a generic sense, and they introduced the algorithm based on tensor decompositions. Typo in Section 2.2: "The techniques of Bhaskara et al. can be applied.." --> ".. can not be applied ..."

Reviewer 2



The paper addresses the problem of learning overcomplete HMM’s with discrete observation space. Main contribution of the paper is a series of necessary conditions that ensure the learnability of the model parameters with time window of finite size. In almost all cases (according to [15]), the length of the minimal time window grows logarithmically with the size of the hidden space. The authors identify a set of transition matrices for which the model is actually learnable with polynomial sample complexity, under certain condition on the observation distributions. Here are few comments: - the paper seems to be closely related to [15], where the same technique is used to prove a quite general result. The two main theorems in the paper refine this result in the following ways: Theorem 1 defines a set of models that belong to the measure-zero set of non-learnable HMM mentioned in [15]; Theorem 2 defines a set models that are learnable under certain conditions on the transition and observation matrices. - the stated necessary conditions are pretty technical and their relevance for practical applications is very shortly discussed. As the major practical application seems to be to language generative models, it would help to see and explicit example or some empirical results on this field. - the applicability of the results relies on some general conditions imposed to the observation matrix through the paper. How `realistic' is assumption 4 on the output distribution? How is it related to the Kruskal rank of the observation matrix? - all results in the paper are obtained by assuming that the HMM is learnt by the method of moments. What can be said about other approaches? Does the proposed conditions apply if the model parameters are obtained via an alternative method? - the paper contains lot of wordy explanations of the technical assumptions and results but any informal description on how the theorems are proven. - in theorem 2, it would be nice to see the explicit dependence of the sample complexity in terms of the singular value of the transition matrix. Is this related to epsilon? In condition 1, the smallest singular value of the the transition matrix is required to be larger than an inverse function of the alphabet size. For very large hidden space this requires c >> 1 and makes the conditions quite restrictive. Are the `allowed' transition matrices still nontrivial in the limit n >> m? - it would be helpful to see a schematic version of the mentioned learning algorithm. What happens if the joint diagonalization approach is replaced by a different tensor decomposition scheme? Some of them apply directly to the overcomplete case, may this allow shorter time windows? - the authors could also add some more extended experiments to show how sparsity, degree and short cycles effects the recovery of the true parameters via the method of moments.